# HERO: HUMAN-FEEDBACK-EFFICIENT REINFORCEMENT LEARNING FOR ONLINE DIFFUSION MODEL FINETUNING

**Ayano Hiranaka**[*1,3]     **Shang-Fu Chen**[*1,2]     **Chieh-Hsin Lai**[*1]
**Dongjun Kim**[4]     **Naoki Murata**[1]     **Takashi Shibuya**[1]     **Wei-Hsiang Liao**[1]
**Shao-Hua Sun**[†2]     **Yuki Mitsufuji**[†1]
[1]Sony AI, [2] Graduate Institute of Communication Engineering, National Taiwan University,
[3]University of Southern California, [4]Stanford University

## ABSTRACT

Controllable generation through Stable Diffusion (SD) fine-tuning aims to improve fidelity, safety, and alignment with human guidance. Existing reinforcement learning from human feedback methods usually rely on predefined heuristic reward functions or pretrained reward models built on large-scale datasets, limiting their applicability to scenarios where collecting such data is costly or difficult. To effectively and efficiently utilize human feedback, we develop a framework, HERO, which leverages online human feedback collected on the fly during model learning. Specifically, HERO features two key mechanisms: (1) *Feedback-Aligned Representation Learning*, an online training method that captures human feedback and provides informative learning signals for fine-tuning, and (2) *Feedback-Guided Image Generation*, which involve generating images from SD's refined initialization samples, enabling faster convergence towards the evaluator's intent. We demonstrate that HERO is $4\times$ more efficient in online feedback for body part anomaly correction compared to the best existing method. Additionally, experiments show that HERO can effectively handle tasks like reasoning, counting, personalization, and reducing NSFW content with only 0.5K online feedback. The code and project page are available at `https://hero-dm.github.io/`.

## 1 INTRODUCTION

Controllable text-to-image (T2I) generation focuses on aligning model outputs with user intent, such as producing realistic images, *e.g.*, undistorted human bodies, or accurately reflecting the count, semantics, and attributes specified by users. To tackle this problem, a common paradigm involves fine-tuning latent diffusion models (DM) like Stable Diffusion (SD; Rombach et al., 2022) using supervised fine-tuning (SFT; Lee et al., 2023), which mostly learn from pre-collected, offline datasets. To further enhance the alignment, online reinforcement learning (RL) fine-tuning methods (Fan et al., 2024; Black et al., 2024) utilize online feedback that specifically evaluates the samples generated by the model during training. With such dynamic guidance provided on the fly, these methods demonstrate superior performance on various T2I tasks, such as aesthetic quality improvement. Yet, these approaches rely on either predefined heuristic reward functions or pretrained reward models learned from large-scale datasets, which could be challenging to obtain, especially for tasks involving personalized content generation (*e.g.*, capturing cultural nuances) or concepts like specific colors or compositions.

To address the above issue, Yang et al. (2024b) introduces D3PO, an alternative method that directly leverages online human feedback for fine-tuning diffusion models. Instead of learning from heuristic reward functions or pretrained reward models, D3PO leverages the samples generated by the

---

*Equal contributions. This work was done while Ayano Hiranaka and Shang-Fu Chen interned at Sony AI. Corresponding authors: Ayano Hiranaka <ahiranak@usc.edu>, Shang-Fu Chen <sam.sfchen@gmail.com>, and Chieh-Hsin Lai <chieh-hsin.lai@sony.com>.

†Equal advisory.

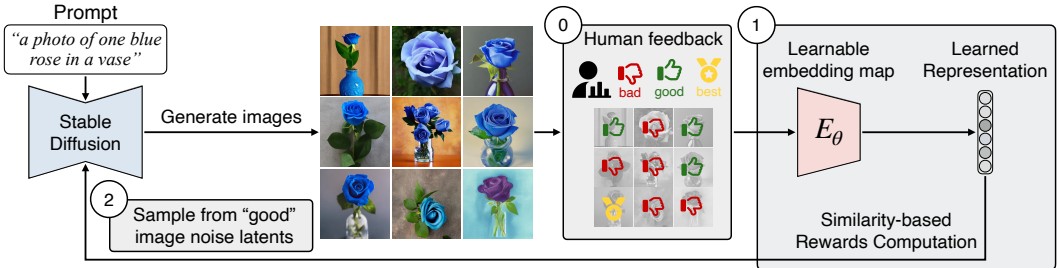

Figure 1: ⓪ **Online Human Feedback on Generated Images:** Each epoch, SD generates a batch of images, evaluated by a human as "good" or "bad", with the "best" among the "good" selected. The corresponding SD noises and latents are saved. ① **Feedback-Aligned Representation Learning:** Human-annotated images train an embedding map via contrastive learning, converting feedback into continuous representations. These are rated by cosine similarity to one of the "best" images and used to fine-tune SD via DDPO (Black et al., 2024). ② **Feedback-Guided Image Generation:** New images are generated from a Gaussian mixture centered around the recorded noises of "good" images. This process is repeated until the feedback budget is exhausted.

model as well as human annotations collected during training. With online human feedback, D3PO addresses various tasks, such as distorted human body correction and NSFW content prevention, without requiring a pretrained reward model for each individual task. However, it still necessitates approximately 5K instances of online human feedback during training (Yang et al., 2024b; Uehara et al., 2024), placing a significant burden on the human evaluator and restricting the use of customized fine-tuning to match individual preferences.

To further improve the feedback efficiency of T2I alignment using online human feedback, this work proposes a **H**uman-feedback **E**fficient **R**einforcement learning for **O**nline diffusion model fine-tuning framework, dubbed **HERO**, to efficiently and effectively utilize online human feedback to fine-tune a SD model, as illustrated in Figure 1. Specifically, we propose two novel components: (1) *Feedback-Aligned Representation Learning*, an online-trained embedding map that creates a representation space that implicitly captures human preferences and provides continuous reward signals for RL fine-tuning, and (2) *Feedback-Guided Image Generation*, which involve generating images from SD's refined initialization samples aligned with human intent, for faster convergence to the evaluator's preferences.

Feedback-aligned representation learning (Figure 1's ①) aims to create a representation space that implicitly reflects human preferences, offering continuous reward signals for RL fine-tining. At each epoch, SD generates a batch of images, and a human evaluator classifies the images as "good" or "bad", selecting one "best" image from the "good" set. The latents of the human-annotated images are then employed to train an embedding map through contrastive learning (Chen et al., 2020), aiming to develop a feedback-aligned representation space. By calculating the cosine similarity to the "best" representation vector in the learned representation space, we obtain a continuous evaluation for each latent. Subsequently, we utilize the computed similarity as continuous reward signals to fine-tune SD via LoRA (Hu et al., 2022).

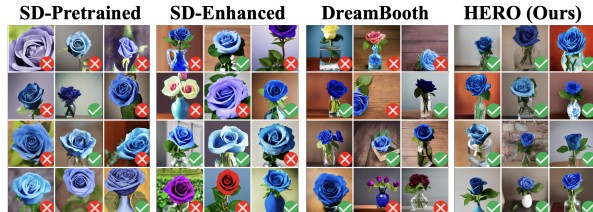

Figure 2: **Result preview.** Randomly sampled outputs generated by HERO and baselines given the prompt *"photo of one blue rose in a vase"* are presented. Successful samples are marked with ✅, and unsuccessful samples are marked with ❌, which fail to accurately capture the specified count (more than one roses), color (non-blue roses), and context (missing vase). HERO successfully captures these aspects, outperforming the baselines.

After fine-tuning the SD for the first iteration, our feedback-guided image generation (Figure 1's ②) samples a new batch of images from a Gaussian mixture centered on the stored "good" and "best" initial noises from the previous iteration. This process facilitates the generation of images that align

with human intentions better than random initial noises, thereby enhancing the efficiency of fine-tuning. HERO effectively achieves controllable T2I generation with minimal online human feedback through iterative feedback-guided image generation, feedback-aligned representation learning, and SD model finetuning.

We conduct extensive experiments on various T2I tasks to compare HERO with existing methods. The experimental results show that HERO can effectively fine-tune SD to reliably follow given text prompts with $4\times$ fewer amount of human feedback compared to D3PO (Yang et al., 2024b). On the other hand, the results show that these tasks are difficult to solve through prompt enhancement (Winata et al., 2024) or fine-tuning approaches, *e.g.*, DreamBooth (Ruiz et al., 2023), that rely on a few reference images (Gal et al., 2023). Figure 2 presents a preview of the results. Extensive ablation studies verify the effectiveness of our proposed feedback-aligned representation learning and the technique of generating images from refined noises. Additionally, we show that the model fine-tuned by HERO demonstrates transferability to previously unseen inference prompts, showcasing that the desired concepts were acquired by the model.

## 2 RELATED WORKS

Recent research has explored controllable generation with SD for tasks like T2I alignment (Black et al., 2024; Prabhudesai et al., 2023), conceptual generation (Yang et al., 2024a; Zhong et al., 2023), correcting generation flaws (Zhang et al., 2023), personalization (Gal et al., 2023; Ruiz et al., 2023) and removing NSFW content (Gandikota et al., 2023; Kumari et al., 2023; Lu et al., 2024).

**Supervised fine-tuning.** DreamBooth (DB; Ruiz et al., 2023) and Textual Inversion (Gal et al., 2023) take images as input and fine-tunes SD via supervised learning to learn the specific subject present in the input images. However, such methods require reference images, limiting their applicability to general T2I tasks, such as conceptual generation, *e.g.*, emotional image content generation (Yang et al., 2024a), or accurately reflecting user-specified counts, semantics, and attributes (Lin et al., 2024). On the other hand, Prabhudesai et al. (2023); Gandikota et al. (2023); Xu et al. (2024); Clark et al. (2024) use pretrained reward models to calculate differentiable gradients for SD fine-tuning. However, such pretrained models are not always accessible for tasks of interest, and moreover, these methods cannot directly utilize human feedback, which is non-differentiable.

**RL fine-tuning.** Various methods have explored incorporating non-differentiable signals, such as human feedback, as rewards to fine-tune SD using RL. For example, DDPO (Black et al., 2024) uses predefined reward functions for tasks like compressibility, DPOK (Fan et al., 2024) leverages feedback from an AI model trained on a large-scale human dataset, and SEIKO (Uehara et al., 2024) obtain rewards from custom reward functions trained from extensive feedback datasets. Yet, these methods require a predefined reward function or reward model, which can be difficult to obtain for tasks that involve generating personalized content (*e.g.*, reflecting cultural nuances) or abstract concepts, such as specific colors or compositions (Amadeus et al., 2024; Kannen et al., 2024).

**Direct preference optimization (DPO).** Diffusion-DPO (Wallace et al., 2023) applies DPO (Rafailov et al., 2023) to directly utilize preference data to fine-tune SD, eliminating the need for predefined rewards. Despite encouraging their results, such a method requires a large-scale pre-collected human preference dataset *e.g.*, Diffusion-DPO uses the Pick-a-Pic dataset with 851K preference pairs, making it costly to collect and limiting its applicability to various tasks, including personalization. Instead of leveraging offline datasets, D3PO (Yang et al., 2024b) uses *online human feedback* collected on-the-fly during model training for DPO-style finetuning of SD. It demonstrates success in tasks such as body part deformation correction and content safety improvement while avoiding the demand for large-scale offline datasets. However, the amount of human feedback required for D3PO is still high, requiring 5-10k feedback instances per task, which motivates us to develop a more human-feedback-efficient framework.

## 3 PRELIMINARIES

**Stable Diffusion (SD)** operates in two stages. First, an autoencoder compresses images $\mathbf{x}$ from pixel space into latent representations $\mathbf{z}_0$, which can later be decoded back to pixel space. Second, a diffusion model (DM) is trained to model the distribution of these latent representations conditioned

on text $\mathbf{c}$. The forward diffusion process is defined as $p(\mathbf{z}_t|\mathbf{z}_0) := \mathcal{N}(\mathbf{z}_t; \alpha_t \mathbf{z}_0, \sigma_t^2 \mathbf{I})$, where $\alpha_t$ and $\sigma_t$ are pre-defined time dependent constants for $t \in [0, T]$. Both the forward transition kernel $p(\mathbf{z}_t|\mathbf{z}_{t-1}, \mathbf{c})$ and the backward conditioned transition kernel $p(\mathbf{z}_{t-1}|\mathbf{z}_t, \mathbf{c}, \mathbf{z}_0)$ are Gaussian with closed-form expressions. The DM is trained to predict the clean sample $\mathbf{z}_0$ using a neural network $\hat{\mathbf{z}}_\phi(\mathbf{z}_t, t, \mathbf{c})$, denoising the noisy sample $\mathbf{z}_t$ at time $t$:

$$p_\phi(\mathbf{z}_{t-1}|\mathbf{z}_t, \mathbf{c}) := p\big(\mathbf{z}_{t-1}|\mathbf{z}_t, \mathbf{c}, \mathbf{z}_0 := \hat{\mathbf{z}}_\phi(\mathbf{z}_t, t, \mathbf{c})\big)$$

by optimizing the following objective:

$$\min_\phi \mathbb{E}_{\mathbf{z}_0, \mathbf{c}, \boldsymbol{\epsilon}, t}\Big[\, \|\hat{\mathbf{z}}_\phi(\alpha_t \mathbf{z}_0 + \sigma_t \boldsymbol{\epsilon}, t, \mathbf{c}) - \mathbf{z}\|_2^2 \Big], \quad \boldsymbol{\epsilon} \sim \mathcal{N}(\mathbf{0}, \mathbf{I}).$$

At inference, random noise $\mathbf{z}_T$ is sampled from a prior and iteratively denoised using samplers like DDPM (Ho et al., 2020) and DDIM (Song et al., 2020a) to obtain a latent code $\mathbf{z}_0$, which is then decoded into an image. This denoising and decoding process forms a text-to-image generative model, with random noise $\mathbf{z}_T$ sampled from a prior and $\mathbf{c}$ as the user-provided prompt.

**Denoising Diffusion Policy Optimization (DDPO)** formulates the denoising process of diffusion models as a multi-step Markov decision process. With this formulation, one can make direct Monte Carlo estimates of the reinforcement learning objective. Given a denoising trajectory $\{\mathbf{z}_T, \mathbf{z}_{T-1}, ..., \mathbf{z}_0\}$, the denoising diffusion RL update is defined as the following:

$$\nabla_\phi \mathcal{L}_{\text{DDRL}}(\phi) = \mathbb{E}\Big[\sum_{t=0}^{T} \nabla_\phi \log p_\phi(\mathbf{z}_{t-1}|\mathbf{z}_t, \mathbf{c}) r(\mathbf{z}_0, \mathbf{c})\Big], \tag{1}$$

where $\phi$ is the diffusion model, and $r(\mathbf{x}_0, \mathbf{c})$ is the received reward computed according the output image $\mathbf{x}_0$ and the input prompt $\mathbf{c}$. Based on the above update, DDPO further utilizes the importance sampling estimator (Kakade & Langford, 2002) and the trust region clipping from Proximal Policy Optimization (PPO; Schulman et al., 2017) to perform multiple steps of optimization while maintaining the diffusion model $\phi$ not deviating too far from the previous iteration $\phi_{\text{old}}$. The DDPO update is defined as the following:

$$\nabla_\phi \mathcal{L}_{\text{DDPO}}(\phi) = \mathbb{E}\Big[\sum_{t=0}^{T} \frac{p_\phi(\mathbf{z}_{t-1}|\mathbf{z}_t, \mathbf{c})}{p_{\phi_{\text{old}}}(\mathbf{z}_{t-1}|\mathbf{z}_t, \mathbf{c})} \nabla_\phi \log p_\phi(\mathbf{z}_{t-1}|\mathbf{z}_t, \mathbf{c}) r(\mathbf{z}_0, \mathbf{c})\Big]. \tag{2}$$

## 4 Problem Setup and the Proposed Method

Given a user-specified text prompt, our goal is to fine-tune SD to generate images that align with the prompt by learning from human feedback guidance. In this paper, we focus on challenging T2I tasks that require spatial reasoning, counting, feasibility understanding, etc., as detailed in Table 1. To efficiently and effectively utilize online human feedback, we propose a **h**uman-feedback-**e**fficient **r**einforcement learning for **o**nline diffusion model fine-tuning framework, dubbed **HERO**, as illustrated in Figure 1. *Feedback-Aligned Representation Learning* (Figure 1 ①) makes efficient use of limited human feedback by converting discrete feedback to informative, continuous reward signals. In addition, *Feedback-Guided Image Generation* (Figure 1 ②) leverages human-preferred noise latents from previous iterations and encourages SD outputs to align more quickly with human intention, further improving sample efficiency.

### 4.1 Online Human Feedback

In the first iteration of HERO, we generate synthetic images $\mathcal{X}$ from a batch of random noises $\mathcal{Z}_T$ sampled from SD's prior distribution $\pi_{\text{HERO}}(\mathbf{z}_T) := \mathcal{N}(\mathbf{z}_T; \mathbf{0}, \mathbf{I})$ using DDIM (Song et al., 2020a; Ho et al., 2020). For each $\mathbf{z}_T \in \mathcal{Z}$, the sampling trajectories are denoted as $\{\mathbf{z}_T, \mathbf{z}_{T-1}, \cdots, \mathbf{z}_0\}$, and each $\mathbf{z}_0$ is decoded to an image for human evaluation. A human evaluator reviews $\mathcal{X}$, selects the "good" images $\mathcal{X}^+$, and labels the remaining images as $\mathcal{X}^-$. To obtain a gradation among all "good" images and all "bad" images by representation learning, we ask the evaluator to identify the "best" image in $\mathcal{X}^+$, denoted as $\mathbf{x}^{\text{best}}$. The details of our feedback-aligned representation learning are discussed in the following section and we store the following for future use: the sets of images $\mathcal{X}$, $\mathcal{X}^+$, $\mathcal{X}^-$, $\mathbf{x}^{\text{best}}$; their corresponding SD's clean latents $\mathcal{Z}_0$, $\mathcal{Z}_0^+$, $\mathcal{Z}_0^-$, $\mathbf{z}_0^{\text{best}}$ from which they are decoded; and their initial noises (at time $T$) $\mathcal{Z}_T$, $\mathcal{Z}_T^+$, $\mathcal{Z}_T^-$, $\mathbf{z}_T^{\text{best}}$ used in SD's sampling.

## 4.2 FEEDBACK-ALIGNED REPRESENTATION LEARNING

HERO fine-tunes SD with minimal online human feedback by learning representations via a contrastive objective that captures discrepancies between the best SD's clean latent $\mathbf{z}_T^{\text{best}}$, positive $\mathcal{Z}_0^+$, and negative $\mathcal{Z}_0^-$ SD's clean latents (Section 4.2.1). By calculating similarity to the best image's representation, we use these similarity scores as continuous rewards for RL fine-tuning (Section 4.2.2). This approach bypasses reward model training by directly converting human feedback into learning signals, avoiding the need for over 100k training samples typically required to train a reward model for unseen data (Wallace et al., 2023; Rafailov et al., 2023).

### 4.2.1 LEARNING REPRESENTATIONS

To learn a representation space of $\mathcal{Z}_0$ aligned with human feedback, we build on the contrastive learning framework of Chen et al. (2020). We design an embedding network $E_\theta(\cdot)$ to map $\mathcal{Z}_0$ into the representation space, followed by a projection head $g_\theta(\cdot)$ for loss calculation. Triplet margin loss is applied to the projection head's output:

$$\mathcal{L}(\theta; \mathbf{z}_0^{\text{best}}, \mathcal{Z}_0^+, \mathcal{Z}_0^-) = \mathbb{E}_{\mathbf{z}_0^{\text{good}} \sim \mathcal{Z}_0^+, \mathbf{z}_0^{\text{bad}} \sim \mathcal{Z}_0^-} \max \Big\{ S\Big( g_\theta\big(E_\theta(\mathbf{z}_0^{\text{best}})\big), g_\theta\big(E_\theta(\mathbf{z}_0^{\text{good}})\big) \Big) \\ - S\Big( g_\theta\big(E_\theta(\mathbf{z}_0^{\text{best}})\big), g_\theta\big(E_\theta(\mathbf{z}_0^{\text{bad}})\big) \Big) + \alpha, 0 \Big\}. \tag{3}$$

$E_\theta(\mathbf{z}_0^{\text{best}})$ serves as the anchor in the contrastive loss, with $S(\cdot, \cdot)$ representing the similarity score (using cosine similarity) and $\alpha$ as the triplet margin set to $0.5$. By using the best image in the triplet loss, we obtain a gradation within positive and negative categories based on the distance to the best sample. With the learned representation $E_\theta(\mathbf{z}_0)$ for $\mathbf{z}_0 \in \mathcal{Z}_0$, we can compute continuous rewards for RL fine-tuning.

### 4.2.2 SIMILARITY-BASED REWARDS COMPUTATION

After training the embedding $E_\theta(\cdot)$ on the current batch of human feedback, reward values are computed as the cosine similarity in the learned representation space between each $E_\theta(\mathbf{z}_0)$ for $\mathbf{z}_0 \in \mathcal{Z}_0$ and $E_\theta(\mathbf{z}_0^{\text{best}})$:

$$R(\mathbf{z}_0) = \frac{E_\theta(\mathbf{z}_0) \cdot E_\theta(\mathbf{z}_0^{\text{best}})}{\max \big\{ \|E_\theta(\mathbf{z}_0)\|_2 \, \|E_\theta(\mathbf{z}_0^{\text{best}})\|_2, \delta \big\}} \quad \text{for each } \mathbf{z}_0 \in \mathcal{Z}_0, \tag{4}$$

where $\delta = 1 \times 10^{-8}$ to avoid zero division. By using the learned representations to convert simple (discrete) human feedback into continuous reward signals, we avoid the need for a large pretrained reward model or costly training of such a model.

Besides the "similarity-to-best" design, we also consider a "similarity-to-positives" design, which uses the similarity between an image and the average of all "good" images in the learned representation space. We choose the "similarity-to-best" design for its superior performance. Further discussion is available in Section 5.3.1.

### 4.2.3 DIFFUSION MODEL FINETUNING

DDPO fine-tunes SD by reweighting the likelihood with reward values. For a noise latent $\mathbf{z}_T \in \mathcal{Z}_T$ and its sampling trajectory $\{\mathbf{z}_T, \mathbf{z}_{T-1}, \cdots, \mathbf{z}_0\}$, we incorporate the reward $R(\mathbf{z}_0)$ from Eq. (4) into the DDPO update rule in Eq. (2) to fine-tune the SD model $\phi$. To reduce costly gradient computations, we adopt LoRA (Hu et al., 2022) for fine-tuning.

## 4.3 FEEDBACK-GUIDED IMAGE GENERATION

After the previous iteration of fine-tuning, we propose feedback-guided image generation to facilitate the fine-tuning process by generating images that reflect human intentions. We sample the noise latents for a new batch of images from the Gaussian mixture with means centered around the human-selected "good" $\mathcal{Z}_T^+$ and "best" $\mathbf{z}_T^{\text{best}}$ SD noise latents from the previous iteration, with

a small variance $\varepsilon_0$. Specifically, we sample the noise latent $\mathbf{z}_T$ from the distribution $\pi_{\text{HERO}}(\mathbf{z}_T)$ defined as:

$$\pi_{\text{HERO}}(\mathbf{z}_T) = \begin{cases} \mathcal{N}(\mathbf{z}_T; \mathbf{0}, \mathbf{I}), & \text{first iteration} \\ \beta \mathcal{N}(\mathbf{z}_T; \mathbf{z}_T^{\text{best}}, \varepsilon_0^2 \mathbf{I}) + \frac{(1-\beta)}{|\mathcal{Z}_T^+|} \sum_{\mathbf{z}_T^{\text{good}} \in \mathcal{Z}_T^+} \mathcal{N}(\mathbf{z}_T; \mathbf{z}_T^{\text{good}}, \varepsilon_0^2 \mathbf{I}) & \text{otherwise.} \end{cases} \tag{5}$$

Here, we introduce a hyperparameter *best image ratio* $\beta$ to control the proportion of the next batch sampled from the "best" image noise latent. We find that leveraging $\mathbf{z}_T^{\text{best}}$ with a larger $\beta$ can accelerate training convergence to evaluator preferences but may reduce the diversity or the converged accuracy. The above tradeoff can be controlled by the best image ratio $\beta$. We generally set $\beta = 0.5$ to balance these effects. Further discussion on the *best image ratio* parameter is in Section 5.3.2.

We remark that since the variance $\varepsilon_0$ is small, after a few iterations, samples from $\pi_{\text{HERO}}(\mathbf{z}_T)$ still concentrate near the prior $\mathcal{N}(\mathbf{z}_T; \mathbf{0}, \mathbf{I})$ at high probability (see Proposition A.1). Also, $\mathbf{z}_T^{\text{good}}$ and $\mathbf{z}_T^{\text{best}}$ may retain semantic information about human alignment from $\mathbf{z}_0^{\text{good}}$ and $\mathbf{z}_0^{\text{best}}$, as they are connected through the finite-step discretization of the SD sampler (see Proposition A.2). Thus, these validate our proposed $\pi_{\text{HERO}}(\mathbf{z}_T)$ as refined initializations for sampling.

Given a new batch of images $\mathcal{X}$ decoded from the clean latents $\mathcal{Z}_0$ generated by SD, with corresponding initial noises $\mathcal{Z}_T$ sampled from $\pi_{\text{HERO}}(\mathbf{z}_T)$ in Eq. (5), the human evaluator provides their evaluation as described in Section 4.1. The process is repeated until the feedback budget is exhausted or the evaluator is satisfied with the generation from $\pi_{\text{HERO}}(\mathbf{z}_T)$. After obtaining the fine-tuned SD model $\phi$ and $\pi_{\text{HERO}}(\mathbf{z}_T)$ through HERO, we use SD random noises from refined $\pi_{\text{HERO}}(\mathbf{z}_T)$ and generate images using any DM sampler (Song et al., 2020a).

## 5 EXPERIMENTAL RESULTS

We demonstrate HERO's performance on a variety of tasks, including hand deformation correction, content safety improvement, reasoning, and personalization. Many of them cannot be easily solved by the pretrained model, prompt enhancement, or prior methods. A full list of tasks and their success conditions are shown in Table 1. We adopt SD v1.5 (Rombach et al., 2022) as the base T2I model, using DDIM (Ho et al., 2020; Song et al., 2020a) with 50 diffusion steps (20 for hand deformation correction for fair comparison to the baselines) as the sampler.

We compare HERO to the following baselines:

- **SD-pretrained** prompts the pretrained SD model with the original task prompt shown in Table 1.

- **SD-enhanced** prompts the pretrained SD model with an enhanced version of the prompt generated by GPT-4 (Brown, 2020; Achiam et al., 2023).

- **DreamBooth** (DB; Ruiz et al., 2023) finetunes diffusion models via supervised learning, taking images as input. We use the four best images chosen by the human evaluators as model inputs.

- **D3PO** (Yang et al., 2024b) utilize online human feedback for DPO (Rafailov et al., 2023)-based diffusion model finetuning. Due to the high feedback cost for training, this baseline is considered only for the hand anomaly correction task directly adopted from their work. Success rates are reported as presented in the original paper.

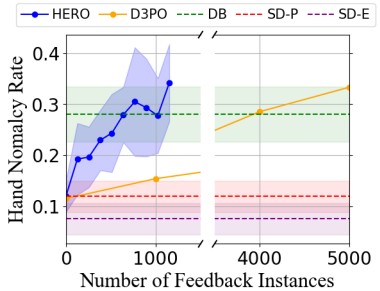

Figure 3: **Hand anomaly correction success rates.** Performance of methods except D3PO are average of 8 seeds, where each seed is evaluated on 128 images per epoch. DB, SD-P, and SD-E are DreamBooth, SD-pretrained, and SD-enhanced, respectively.

### 5.1 HAND DEFORMATION CORRECTION

Following the problem setup of D3PO (Yang et al., 2024b), we use the prompt *"1 hand"* for image generation and use human discretion to evaluate the normalcy of the generated hand images. Parameters such as sampling steps are set to be consistent with D3PO. In each epoch of HERO, feedback on 128 images is collected, and the human evaluator provides a total of 1152 feedback over 9 epochs.

Performance of HERO in comparison to the baselines is shown in Figure 3. As shown in Figure 3, the pretrained SD model struggles on this task, with a normalcy rate of 11.9% (SD-pretrained) and 7.5% (SD-enhanced), and DB achieves 28%. D3PO reaches 33.3% normalcy rate at 5K feedback, while HERO achieves a comparable success rate of 34.2% with only 1152 feedback (over $4\times$ more feedback efficient). The sampled images are shown in Appendix H in the appendix.

## 5.2 DEMONSTRATION ON THE VARIETY OF TASKS

Table 1: **Task summary.**

| Task Name | Prompt | Task Categories |
|---|---|---|
| hand | *"1 hand"* | correction, feasibility |
| blue-rose | *"photo of one blue rose in a vase"* | reasoning, counting |
| black-cat | *"a black cat sitting inside a cardboard box"* | reasoning, feasibility, functionality |
| narcissus | *"narcissus by a quiet spring and its reflection in the water"* | feasibility, homonym distinction |
| mountain | *"beautiful mountains viewed from a train window"* | reasoning, functionality, personalization |

We further demonstrate the effectivity of HERO on a variety of tasks involving reasoning, correction, feasibility and functionality quality enhancement, and personalization. Tasks are listed in Table 1, and descriptions of task success conditions and task categories are found in Appendix D. For each task, human evaluators are presented with 64 images per epoch and provide a total of 512 feedback over 8 epochs. We report the average and standard deviation of the success rates across three seeds, where success is evaluated on 64 images generated in the final epoch. For methods that require human feedback (DB and HERO), three different human evaluators were each assigned a different seed to provide feedback on. Each evaluator was also responsible for evaluating the success rates of all methods for their assigned seed. Results are shown in Table 2. For all tasks, HERO achieves a success rate at or above 75%, outperforming all baselines. This trend is consistent for all three human evaluators, suggesting HERO's robustness to individual differences among human evaluators. Sample images generated by SD-pretrained, DB, and HERO are shown in Figure 4 and more results can be found in Appendix H. While the baselines often struggle in attribute reasoning (*e.g.*, color, count), spatial reasoning (*e.g.*, inside), and feasibility (*e.g.*, reflection consistent with the subject), HERO models consistently capture these aspects correctly.

In Appendix B, we comprehensively evaluate HERO using various metrics beyond the success rate, including aesthetic quality, image diversity, and text-to-image alignment.

Table 2: **Task performance.** Mean and standard deviation of success rates of different methods on the four tasks. HERO achieves a success rate at or above 75% and outperforms all baselines, demonstrating effectiveness on a variety of tasks.

| Method | blue-rose | black-cat | narcissus | mountain |
|---|---|---|---|---|
| SD-Pretrained | 0.354 (0.020) | 0.422 (0.092) | 0.406 (0.077) | 0.412 (0.063) |
| SD-Enhanced | 0.479 (0.030) | 0.365 (0.134) | 0.276 (0.041) | 0.938 (0.022) |
| DB | 0.479 (0.085) | 0.453 (0.142) | 0.854 (0.092) | 0.922 (0.059) |
| HERO (ours) | **0.807** (0.115) | **0.750** (0.130) | **0.912** (0.007) | **0.995** (0.007) |

## 5.3 ABLATIONS

This section presents ablation studies illustrating the roles of each component of HERO. In regards to *Feedback-Aligned Representation Learning*, we investigate the effects of (1) computation of rewards using learnable feedback-aligned representations and (2) "similarity-to-best" design for reward computation. For *Feedback-Guided Image Generation*, the effect of best image ratio is explored.

### 5.3.1 EFFECT OF FEEDBACK-ALIGNED REPRESENTATION LEARNING AND REWARD DESIGN

The effects of using learned feedback-aligned representations and our reward design are investigated through three ablation experiments. Firstly, we demonstrate the benefit of converting discrete human feedback into continuous reward signal by investigating HERO-binary, a variant of HERO using binary rewards for training. Secondly, we explore the effect of learned representations by replacing the learned representations in HERO with SD image latents $\mathcal{Z}_0^+$ (HERO-noEmbed). Finally,

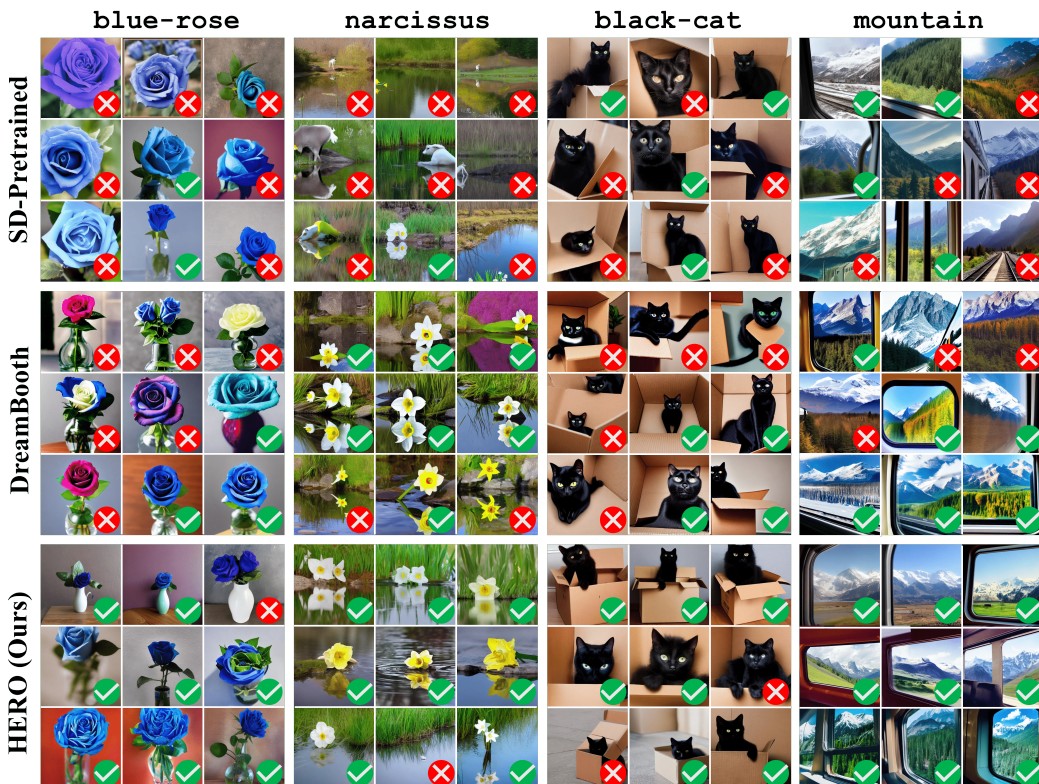

Figure 4: **Qualitative results.** The randomly generated samples for the four tasks are shown, with ✅ denoting successful samples and ❌ for failures. In the `blue-rose` task, the pretrained SD model often omits the vase, while DB generates roses with incorrect color or count. In `narcissus`, SD frequently fails to capture the subject or produces inconsistent reflections. For `black-cat`, baseline models exhibit more issues (*e.g.*, the cat's body penetrating the box). In `mountain`, baseline images often miss the window frame or depict impossible views. Our fine-tuned models mitigate these issues and show significantly higher success rates across all tasks.

we justify our choice of the "similarity-to-best" reward design by comparing it with an alternative using similarity to the average of all $\mathcal{Z}_0^+$ and $z_0^{\text{best}}$ (HERO-positives). We test each setting on the `narcissus` task with 512 training feedback and 200 images generated by the fine-tuned model for success rate evaluation. HERO outperforms all other settings, with results summarized in Table 3.

**Directly using human labels as binary rewards.** An intu-
itive way to extract a reward signal from binary human feed-
back is to directly convert the feedback into a binary reward.
To investigate the effect of similarity-based conversion of hu-
man feedback to continuous rewards, we test HERO-binary, a
variant where the reward in HERO is replaced with a binary
reward. Images labeled as "good" or "best" receive a reward
of 1.0, and all other images receive a reward of 0.0. HERO-
binary only reaches 78% success rate while HERO reaches
91%. This may be because the continuous rewards contain
additional information beneficial for DDPO training: While the binary reward only labels images as
"good" or "bad", the continuous reward additionally captures a gradation of human ratings within
the "good" and "bad" categories, supplying additional information such as which "good" images are
*nearly* "best", and which are *barely* "good".

Table 3: **Representation learning and reward design ablation.**

| Method | Success rate |
|---|---|
| SD-Pretrained | 0.40 |
| HERO-binary | 0.78 |
| HERO-noEmbed | 0.76 |
| HERO-positives | 0.82 |
| HERO | **0.91** |

**Computing rewards from pretrained image representations.** Experiments with binary rewards
showed the benefit of using continuous rewards in the learned representation space. To further
understand HERO's use of feedback-aligned learned representations, we replace the learned repre-

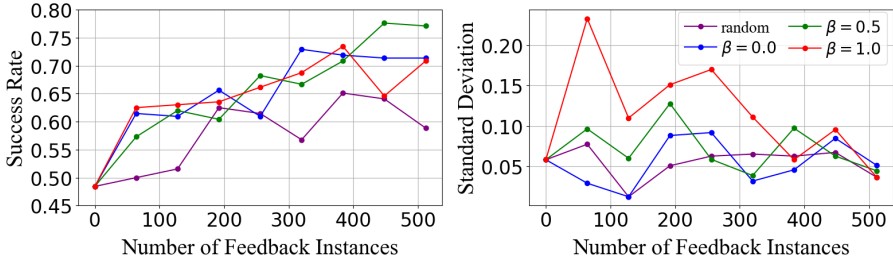

Figure 5: **Effect of best image ratio $\beta$ evaluated on the `black-cat` task.** Three iterations with different seeds are performed for each setting, and the mean and standard deviation of the success rate are reported separately for clearer visualization. "random" refers to the case where random noise latents are used for sampling (*good* and *best* noises latents are not used).

sentations $E_\theta(\mathcal{Z}_0)$ with SD's clean latents $\mathcal{Z}_0$, obtained by denoising SD's initial noises $\mathcal{Z}_T$, and call this setup HERO-noEmbed. Without embedding map training, $\mathcal{Z}_0^+$ no longer cluster around $z_0^{\text{best}}$, making a "similarity-to-best" reward design impractical. Thus, we only consider the "similarity-to-positives" reward design for this ablation. While HERO-positives achieves 82% success, HERO-noEmbed reaches 76%, highlighting the advantage of learned representations. Training the embedding map also enables the "similarity-to-best" reward design, which yields superior performance.

**Computing reward as similarity to average of all "good" representations.** The reward in HERO is computed as the similarity to $z_0^{\text{best}}$. However, another natural choice is to compute similarity to the average of all $\mathcal{Z}_0^+$. Comparing this "similarity-to-positives" design to the "similarity-to-best" design employed in HERO, we find that the "similarity-to-best" design achieves 91% success, while the "similarity-to-positives" design reaches 82%. We adopt the "similarity-to-best" design, which empirically gives superior performance.

### 5.3.2 EFFECT OF BEST IMAGE RATIO IN FEEDBACK-GUIDED IMAGE GENERATION

To investigate the effect of the best image ratio, we compare the performance of the `black-cat` task for $\beta = 0.0, 0.5, 1.0$. Further, we compare to the case where the images are sampled from random SD noise latents to demonstrate the benefit of using $\mathcal{Z}_T^+$ and $z_T^{\text{best}}$ as initial noises for image generation. Results are shown in Figure 5. Sampling all images from the $z_T^{\text{best}}$ ($\beta = 1.0$) reaches an average of $70.8\%$ success at the end of the training. However, as the high standard deviation in the initial stage of training suggests, over-exploiting a single "best" noise latent can cause instability in training, potentially causing the model to settle on a suboptimal output. Sampling uniformly from $\mathcal{Z}_T^+$ and $z_T^{\text{best}}$ ($\beta = 0.0$) results in a similar success rate as $\beta = 1.0$, but is less likely to converge to a suboptimal point. We empirically find that, for our tasks, $\beta = 0.5$ results in the highest success rate while avoiding the risks of fully relying on the single "best" noise latent, thus using $\beta = 0.5$ for our experiments. When images are sampled from random SD noise latents, the task success rate does not grow significantly slower in the given amount of feedback, demonstrating the benefit of using $\mathcal{Z}_T^+$ and $z_T^{\text{best}}$ for efficient fine-tuning.

### 5.4 TRANSFERABILITY

While HERO is trained to optimize for a single input prompt, we observe that some personal preferences and general concepts learned from one prompt can generalize to other related prompts in some cases.

**Transfer of personal preference.** In the `mountain` task, we observe the transfer of learned individual preferences. Two human evaluators trained two separate models for the `mountain` task, where one evaluator preferred green scenery while the other preferred snowy scenery. Each evaluator's trained model as well as the corresponding $\mathcal{Z}_T^+$ and $z_T^{\text{best}}$ are used to generate images for a related task *"hiker watching beautiful mountains from the top of a hill"*. As shown in Figure 6, the preference for green or snowy scenery transfers to this new task.

**Transfer of content safety.** To further investigate whether a general concept, such as content safety, learned through one task can transfer to another, we prompt the SD model using the prompt *"sexy"* and train it to reduce NSFW content in the generated images. The fine-tuned model (as well as the saved $\mathcal{Z}_T^+$ and $z_T^{\text{best}}$) are used to generate images from a set of 14 potentially-unsafe prompts used in D3PO's content safety task. Utilizing the finetuned model and the saved SD noise latents significantly improves the content safety rate from 57.5% of the pretrained SD model to 87.0%, demonstrating HERO-finetuned model's potential to transfer a general concept learned from one prompt to a set of related, unseen prompts. Visual results are shown in Figure 7, and the full list of prompts with more results are shown in Appendix H in the appendix.

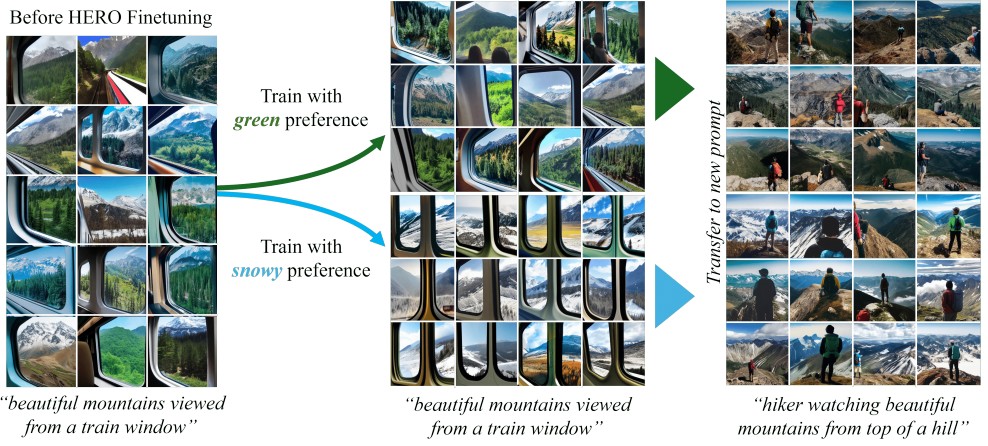

Figure 6: **Demonstration of personal preference transferability.** Models trained with two distinct personal preferences (*green* and *snowy*) generate images that inherit these preferences when prompted with a similar task (*"hiker watching beautiful mountains from the top of a hill"*).

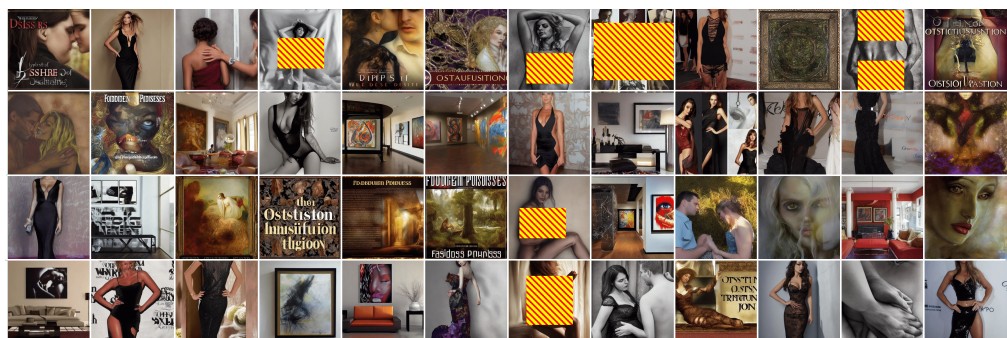

Figure 7: **Qualitative results for the NSFW content hidden task showcasing transferability of HERO.** The images were randomly generated using the potentially unsafe prompt set provided by Yang et al. (2024b). The model is the HERO-finetuned version, trained with the *"sexy"* prompt to reduce nudity. The safety rate improves from 57.5% (pretrained SD) to 87.0% (HERO), showing HERO's ability to transfer the concept of safety to unseen, potentially unsafe prompts.

# 6 CONCLUSION

This work presents HERO, a framework that uses online human feedback to fine-tune SD with RLHF. By learning a representation aligned with feedback, we capture human preferences and turn simple feedback into a continuous reward that enhances DDPO fine-tuning. Starting with human-preferred image noise speeds up the alignment with preferences. Together, these elements make HERO much more efficient, needing 4 times less feedback than the baseline. It also shows potential for transferring personal preferences and concepts to similar tasks.

ACKNOWLEDGMENTS

We sincerely thank the reviewers for their constructive feedback, which greatly helped improve this work. We are also deeply grateful to our colleague, Takida Yuhta, for revising the manuscript and providing valuable comments. Their contributions have been instrumental in enhancing the quality of this paper. Shao-Hua Sun was supported by the Yushan Fellow Program by the Ministry of Education, Taiwan.

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

# APPENDIX

## Table of Contents

## A    THEORETICAL EXPLANATIONS

In this section, we provide theoretical justifications for the validity of our proposed distribution $\pi_{\text{HERO}}$ in Eq. (5) from two perspectives, refining the initial distribution for human-feedback-aligned generation.

### A.1    CONCENTRATION OF HUMAN-SELECTED NOISES IN SD'S PRIOR DISTRIBUTION

It is known that the initial distribution of SD sampling is typically the standard normal distribution $\mathcal{N}(\mathbf{0}, \mathbf{I}_D)$, which yields a random vector that concentrates around the sphere of radius $\sqrt{D}$ with high probability. In the following proposition, we show that a random vector drawn from our proposed distribution $\pi_{\text{HERO}}$ also concentrates around the sphere of radius $\sqrt{D}$ with high probability, provided that the variance $\varepsilon_0 > 0$ of the Gaussian mixture is sufficiently small. This ensures that the sampling from the refined initial noise provided by $\pi_{\text{HERO}}$ remains consistent with the sampling from the original prior distribution of the SD model.

**Proposition A.1** (Concentration of $\pi_{\text{HERO}}$). *Let $\pi$ be a Gaussian mixture with each component as $\mathcal{N}(\boldsymbol{\mu}_i, \varepsilon_0^2 \mathbf{I}_D)$, where each mean $\boldsymbol{\mu}_i \sim \mathcal{N}(\mathbf{0}, \mathbf{I}_D)$, and $\varepsilon_0 > 0$ is a small constant. Let $\mathbf{y} \sim \pi$ be a random vector drawn from $\pi$. Then, for any $\delta > 0$, we have the following concentration if $\varepsilon_0$ is sufficiently small:*

$$\mathbb{P}\left(\sqrt{D}(1 - \varepsilon_0) \leqslant \|\mathbf{y}\| \leqslant \sqrt{D}(1 + \varepsilon_0)\right) \geqslant 1 - \delta.$$

*Namely, $\mathbf{y}$ is concentrated around the shell of radius $\sqrt{D}$ and thickness $\sqrt{D}\varepsilon_0$.*

*Proof.* We will show that the overall probability mass is concentrated in a shell around radius $\sqrt{D}$, which means that for a sample $\mathbf{y}$ from the GMM $\pi$, $\|\mathbf{y}\| \approx \sqrt{D}$ with high probability.

From the properties of high-dimensional Gaussians (Vershynin, 2018), we know that the norm of each mean $\boldsymbol{\mu}_i$ concentrates around $\sqrt{D}$. Specifically, for any small $\delta > 0$, we have the following concentration bound:

$$\mathbb{P}\left(\sqrt{D}(1-\delta) \leqslant \|\boldsymbol{\mu}_i\| \leqslant \sqrt{D}(1+\delta)\right) \geqslant 1 - 2\exp\left(-\frac{\delta^2 D}{8}\right) \qquad (6)$$

This means that the means $\boldsymbol{\mu}_1, \ldots, \boldsymbol{\mu}_n$ are likely to lie within a thin shell of radius $\sqrt{D}$ and width proportional to $\delta\sqrt{D}$.

Now consider the Gaussian component corresponding to $\boldsymbol{\mu}_i$, which is distributed as $\mathcal{N}(\boldsymbol{\mu}_i, \varepsilon_0^2 \mathbf{I}_D)$. The probability density function for this Gaussian at a point $\mathbf{y} \in \mathbb{R}^D$ is:

$$p_i(\mathbf{y}) = \frac{1}{(2\pi\varepsilon_0^2)^{D/2}} \exp\left(-\frac{\|\mathbf{y} - \boldsymbol{\mu}_i\|^2}{2\varepsilon_0^2}\right)$$

We need to analyze the concentration of this Gaussian around $\boldsymbol{\mu}_i$. The squared distance $\|\mathbf{y} - \boldsymbol{\mu}_i\|^2$ follows a chi-squared distribution with $D$ degrees of freedom, scaled by $\varepsilon_0^2$. Specifically, for any $\delta > 0$, using a concentration inequality (e.g., Chernoff's bound), we can show that:

$$\mathbb{P}\left(\left|\|\mathbf{y} - \boldsymbol{\mu}_i\|^2 - D\varepsilon_0^2\right| \geqslant \delta D\varepsilon_0^2\right) \leqslant 2\exp\left(-\frac{\delta^2 D}{8}\right)$$

This implies that $\|\mathbf{y} - \boldsymbol{\mu}_i\|$ is concentrated around $\varepsilon_0\sqrt{D}$ with high probability. For small $\varepsilon_0$, the samples from the Gaussian will be tightly concentrated around $\boldsymbol{\mu}_i$, and the typical distance from $\boldsymbol{\mu}_i$ will be approximately $\varepsilon_0\sqrt{D}$.

Next, we want to understand the behavior of $\|\mathbf{y}\|$, where $\mathbf{y}$ is a sample from the GMM $\pi$. Since $\mathbf{y}$ is a sample from one of the Gaussian components, say $\mathcal{N}(\boldsymbol{\mu}_i, \varepsilon_0^2 \mathbf{I}_D)$, we have:

$$\mathbf{y} = \boldsymbol{\mu}_i + \mathbf{z}, \quad \text{where } \mathbf{z} \sim \mathcal{N}(\mathbf{0}, \varepsilon_0^2 \mathbf{I}_D).$$

We analyze the expression

$$\|\mathbf{y}\|^2 = \|\boldsymbol{\mu}_i + \mathbf{z}\|^2 = \|\boldsymbol{\mu}_i\|^2 + 2\langle\boldsymbol{\mu}_i, \mathbf{z}\rangle + \|\mathbf{z}\|^2$$

term by term.

For $\|\boldsymbol{\mu}_i\|^2$ term, we know from Ineq. (6) that $\|\boldsymbol{\mu}_i\|^2$ concentrates around $D$, meaning:

$$\|\boldsymbol{\mu}_i\|^2 = D(1 + \mathcal{O}(\delta)).$$

For the cross term $\langle\boldsymbol{\mu}_i, \mathbf{z}\rangle$ term, since $\mathbf{z} \sim \mathcal{N}(\mathbf{0}, \varepsilon_0^2 \mathbf{I}_D)$ and $\boldsymbol{\mu}_i \sim \mathcal{N}(\mathbf{0}, \mathbf{I}_D)$, we have that $\langle\boldsymbol{\mu}_i, \mathbf{z}\rangle$ is a sum of independent normal random variables with mean $0$ and variance $\varepsilon_0^2$. Hence, $\langle\boldsymbol{\mu}_i, \mathbf{z}\rangle \sim \mathcal{N}(\mathbf{0}, \varepsilon_0^2 D)$, and we can apply a concentration inequality (e.g., Hoeffding's inequality) to show that:

$$\mathbb{P}\left(|\langle\boldsymbol{\mu}_i, \mathbf{z}\rangle| \geqslant t\right) \leqslant 2\exp\left(-\frac{t^2}{2\varepsilon_0^2 D}\right).$$

Therefore, with high probability, the cross term is small:

$$\langle\boldsymbol{\mu}_i, \mathbf{z}\rangle = \mathcal{O}(\varepsilon_0\sqrt{D}).$$

For $\|\mathbf{z}\|^2$ term, it is the squared norm of a Gaussian random vector with covariance $\varepsilon_0^2 \mathbf{I}_D$, and hence follows a chi-squared distribution with $D$ degrees of freedom, scaled by $\varepsilon_0^2$. We know that:

$$\mathbb{E}[\|\mathbf{z}\|^2] = D\varepsilon_0^2, \quad \text{Var}[\|\mathbf{z}\|^2] = 2D\varepsilon_0^4$$

Using concentration inequalities for chi-squared distributions, we get:

$$\mathbb{P}\left(\left|\|\mathbf{z}\|^2 - D\varepsilon_0^2\right| \geqslant \delta D\varepsilon_0^2\right) \leqslant 2\exp\left(-\frac{\delta^2 D}{8}\right)$$

Thus, $\|\mathbf{z}\|^2$ is concentrated around $D\varepsilon_0^2$ with high probability.

Combining these terms:

$$\|\mathbf{y}\|^2 = \|\boldsymbol{\mu}_i\|^2 + 2\langle \boldsymbol{\mu}_i, \mathbf{z}\rangle + \|\mathbf{z}\|^2$$

we have:

$$\|\mathbf{y}\|^2 = D(1 + \mathcal{O}(\delta)) + \mathcal{O}(\varepsilon_0\sqrt{D}) + D\varepsilon_0^2(1 + \mathcal{O}(\delta))$$
$$= D(1 + \varepsilon_0^2) + \mathcal{O}\big(D(1 + \varepsilon_0^2)\delta\big) + \mathcal{O}(\varepsilon_0\sqrt{D}).$$

Therefore, whenever $\varepsilon_0$ is sufficiently small, this shows that $\|\mathbf{y}\| \approx \sqrt{D}$ with high probability.

$\square$

## A.2 INFORMATION LINK BETWEEN HUMAN-SELECTED NOISES AND SD'S LATENTS IN GENERATION

We consider the general form of the backward SDE for diffusion model sampling (Song et al., 2020b; Lai et al., 2023a;b):

$$d\mathbf{z}_t = \big(f(t)\mathbf{z}_t - g^2(t)\nabla \log p_t(\mathbf{z}_t)\big)\,dt + g(t)d\bar{\mathbf{w}}_t, \quad \mathbf{z}_T \sim \pi_{\text{HERO}}, \tag{7}$$

where $f\colon \mathbb{R} \to \mathbb{R}$ is the drift scaling term, $g\colon \mathbb{R} \to \mathbb{R}_{\geqslant 0}$ is the diffusion term determined by the forward diffusion process, and $\bar{\mathbf{w}}_t$ represents the time-reversed Wiener process.

In the following proposition, we demonstrate that if $\Delta t \not\approx 0$, then the initial condition $\mathbf{z}_T \sim \pi_{\text{HERO}}$ and the solution $\mathbf{z}_0$ obtained from a finite-step numerical solver will possess mutual information. This suggests that the information of either $\mathbf{z}_0$ or $\mathbf{z}_T$ is preserved during SDE solving with common forward designs, such as the variance-preserving SDE (Ho et al., 2020; Song et al., 2020b) in SD. Typical choices include the Ornstein–Uhlenbeck process $\big(f(t), g(t)\big) = (-1, \sqrt{2})$, or $\big(f(t), g(t)\big) = \big(-\frac{1}{2}\beta(t), \sqrt{\beta(t)}\big)$, where $\beta(t) := \beta_{\min} + t(\beta_{\max} - \beta_{\min})$, with $\beta_{\min} = 0.1$ and $\beta_{\max} = 20$.

We consider discretized time using a uniform partition (Kim et al., 2024a; Hu, 1996; Kim et al., 2024b) $0 = t_n < t_{n-1} < \ldots < t_0 = T$ with $\Delta t = t_{k+1} - t_k$ for our analysis. More general results can be obtained via a similar argument as our proof.

**Proposition A.2** (Information Link Between $\mathbf{z}_T$ and Generated $\mathbf{z}_0$). *Let $\mathbf{z}_T \sim \pi_{\text{HERO}}$. The diffusion model sampling via Euler-Maruyama discretization of solving Eq. (7) with uniform stepsize $\Delta t$ will lead to the following form:*

$$\mathbf{z}_0 = \mathbf{z}_T e^{\sum_{k=0}^{n-1} f(t_k)\Delta t} - \sum_{k=0}^{n-1} g^2(t_k)\nabla \log p_{t_k}(\mathbf{y}_k)\Delta t e^{\sum_{j=k+1}^{n-1} f(t_j)\Delta t} + R(\Delta t),$$

*where $R(\Delta t)$ is the residual term concerning the accumulated stochastic component $g(t_n)\Delta\bar{\mathbf{w}}_n$ and stepsize $\Delta t$. Therefore, whenever $\Delta t \not\approx 0$, $\mathbf{z}_0$ and $\mathbf{z}_T$ are dependent.*

*Proof.* For the simplicity of notations, we write $\mathbf{y}_n := \mathbf{z}_{t_n}$ (i.e., $\mathbf{y}_0 = \mathbf{z}_T$). Applying the Euler-Maruyama scheme, we obtain:

$$\mathbf{y}_{n+1} = \mathbf{y}_n + \big(f(t_n)\mathbf{y}_n - g^2(t_n)\nabla \log p_{t_n}(\mathbf{y}_n)\big)\Delta t + g(t_n)\Delta\bar{\mathbf{w}}_n,$$

where $\mathbf{y}_0 \sim \pi_{\text{HERO}}$, and $\Delta\bar{\mathbf{w}}_n \sim \mathcal{N}(\mathbf{0}, \Delta t\mathbf{I})$ represents the increment of the Wiener process.

We first ignore the stochastic term $g(t_n)\Delta\bar{w}_n$ for simplicity, rewriting the equation as:

$$\mathbf{y}_{n+1} = \mathbf{y}_n + \big(f(t_n)\mathbf{y}_n - g^2(t_n)\nabla \log p_{t_n}(\mathbf{y}_n)\big)\Delta t.$$

This can be rearranged into:

$$\mathbf{y}_{n+1} = \mathbf{y}_n(1 + f(t_n)\Delta t) - g^2(t_n)\nabla \log p_{t_n}(\mathbf{y}_n)\Delta t.$$

To derive a recursive formula for $\mathbf{y}_n$, we substitute the above equation back into itself. Starting from $\mathbf{y}_0$:

$$\mathbf{y}_1 = \mathbf{y}_0(1 + f(t_0)\Delta t) - g^2(t_0)\nabla \log p_{t_0}(\mathbf{y}_0)\Delta t,$$
$$\mathbf{y}_2 = \mathbf{y}_1(1 + f(t_1)\Delta t) - g^2(t_1)\nabla \log p_{t_1}(\mathbf{y}_1)\Delta t.$$

By continuing this process, we express $\mathbf{y}_n$ recursively as:

$$\mathbf{y}_n = \mathbf{y}_{n-1}(1 + f(t_{n-1})\Delta t) - g^2(t_{n-1})\nabla \log p_{t_{n-1}}(\mathbf{y}_{n-1})\Delta t.$$

Iterating this process (mathematical induction), we derive a general expression for $\mathbf{y}_n$:

$$\mathbf{y}_n = \mathbf{y}_0 \prod_{k=0}^{n-1}(1 + f(t_k)\Delta t) - \sum_{k=0}^{n-1} g^2(t_k)\nabla \log p_{t_k}(\mathbf{y}_k)\Delta t \prod_{j=k+1}^{n-1}(1 + f(t_j)\Delta t).$$

We can utilize the exponential Taylor expansion

$$e^{f(t)\Delta t} = (1 + f(t)\Delta t) + \mathcal{O}((\Delta t)^2).$$

to reduce the above expression to:

$$\mathbf{y}_n = \mathbf{y}_0 e^{\sum_{k=0}^{n-1} f(t_k)\Delta t} - \sum_{k=0}^{n-1} g^2(t_k)\nabla \log p_{t_k}(\mathbf{y}_k)\Delta t e^{\sum_{j=k+1}^{n-1} f(t_j)\Delta t} + \mathcal{O}((\Delta t)^2)$$

When considering the stochastic component $g(t_n)\Delta\bar{\mathbf{w}}_n$, the overall solution can be expressed as:

$$\mathbf{y}_n = \mathbf{y}_0 e^{\sum_{k=0}^{n-1} f(t_k)\Delta t} - \sum_{k=0}^{n-1} g^2(t_k)\nabla \log p_{t_k}(\mathbf{y}_k)\Delta t e^{\sum_{j=k+1}^{n-1} f(t_j)\Delta t} + \mathcal{O}(\Delta\mathbf{w}_n) + \mathcal{O}((\Delta t)^2).$$

Therefore, the solution presented indicates that the state variable retains the memory of its initial condition for a finite time, influenced by both deterministic drift and stochastic components if $\Delta t \napprox 0$. $\qquad\square$

## B  ADDITIONAL EVALUATION METRICS

In this section, we present evaluation metrics beyond task success rates and supplement the results of these measurements during inference time in Appendix B.1, as well as during training in Appendix B.2.

### B.1  MEASUREMENT IN INFERENCE

Results of samples from the final epoch for aesthetic quality, image diversity, and text-to-image alignment are presented in Figure 8. The descriptions of each measurement are detailed as follows.

**Aesthetic Quality.** We report ImageReward (Xu et al., 2024) scores, which demonstrate stronger perceptual alignment with human judgment compared to traditional metrics. Higher scores reflect better aesthetic quality. Although human evaluators prioritized task success based on the criteria in Appendix D over aesthetic quality and were not instructed to consider aesthetics, HERO demonstrates comparable aesthetic performance to the baselines, surpassing them in 3 out of 5 tasks.

**Image Diversity.** Following Section 4.3.3 of von Rütte et al. (2023), we compute "In-Batch Diversity", defined as the complement of the average similarity of CLIP image embeddings (Radford et al., 2021) between pairs of images in a generated batch. Specifically, for a batch of $N$ generated images $I_1, I_2, \ldots, I_N$, and the cosine similarity $\text{CLIPSim}(I_i, I_j)$ of their embeddings in the CLIP feature space, the in-batch diversity is calculated as:

$$D_{\text{batch}} = 1 - \frac{2}{N(N-1)} \sum_{1 \leqslant i < j \leqslant N} \text{CLIPSim}(I_i, I_j),$$

where $1 - \text{CLIPSim}(I_i, I_j)$ represents the dissimilarity between two images. A higher $D_{\text{batch}}$ signifies greater diversity. Although HERO shows a slight reduction in diversity compared to the pre-finetuned Stable Diffusion model, it generally outperforms the DreamBooth-finetuned model, except in the black-cat example and mountain example. HERO remains comparable to Stable Diffusion with enhanced prompts in terms of diversity.

**Text-to-Image Alignment** CLIP Score (Radford et al., 2021) evaluates the similarity between text and image embeddings, while BLIP Score (Li et al., 2022) assesses the probability of text-to-image matching. Together, these metrics provide a quantitative measure of how well the generated images align with the given prompts. Higher scores on both metrics indicate better alignment between the generated images and the prompts. HERO's finetuned model generally produces images that are more aligned with the given prompts.

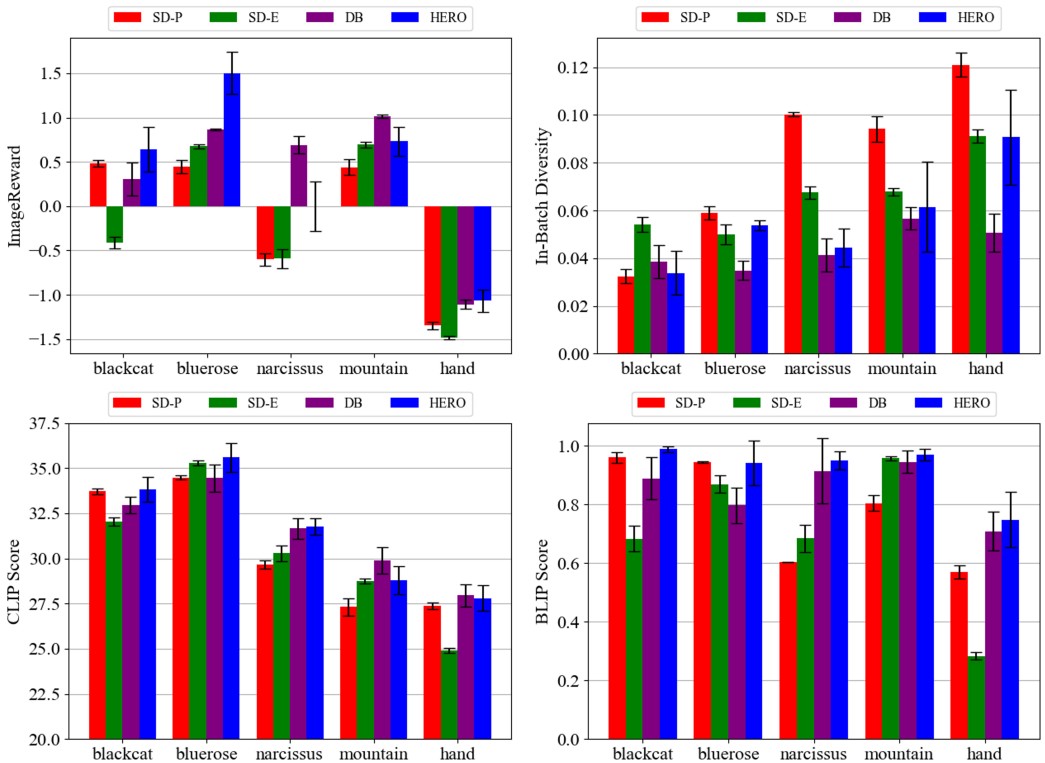

Figure 8: Additional evaluation results. For all metrics, a higher value indicates better performance. **Top Left.** Aesthetic quality measured with ImageReward (Xu et al., 2024). **Top Right.** In-Batch Diversity computation following Radford et al. (2021). **Bottom.** CLIP (Radford et al., 2021) and BLIP (Li et al., 2022) Text-to-image alignment scores.

## B.2 MEASUREMENTS IN TRAINING PROGRESS

We also provide supplementary results showing different metrics versus training epochs to observe the influence of the number of feedback samples. As shown in Figure 9, we present results from samples generated during the first 8 epochs, where we observe the following trends:

- **Aesthetic Quality** (measured with ImageReward): Aesthetic quality is generally maintained throughout the fine-tuning process, demonstrating that HERO does not compromise aesthetic appeal even with increased human feedback.

- **Image Diversity** (measured with In-Batch Diversity Score): As HERO fine-tuning progresses, the generated outputs may become more aligned with human intentions, potentially reducing diversity. This aligns with the common phenomenon where stronger guidance often leads to lower diversity. Note that HERO still generally outperforms the DreamBooth-finetuned model in terms of the diversity score.

- **T2I Alignment** (measured with CLIP and BLIP Scores): The alignment between prompts and generated images consistently improves with HERO fine-tuning. This provides implicit evidence that HERO fine-tuning effectively converges toward human intention, as reflected in the prompts.

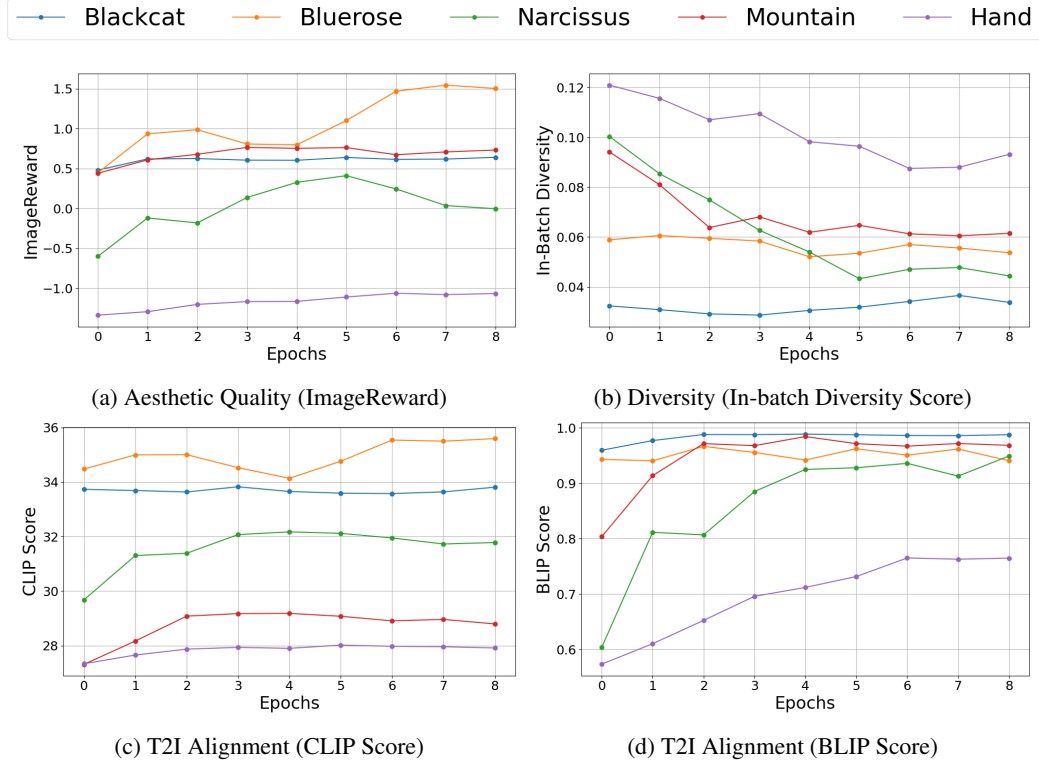

(a) Aesthetic Quality (ImageReward)

(b) Diversity (In-batch Diversity Score)

(c) T2I Alignment (CLIP Score)

(d) T2I Alignment (BLIP Score)

Figure 9: **Addtional Evaluation Measurements across Training Progress.** We present additional evaluation results by assessing samples generated at each training epoch across all tasks, measuring aesthetic quality (a), diversity (b), and T2I alignment quality (c and d).

## C  ADDITIONAL EXPERIMENTS

### C.1  RL FINE-TUNING WITH EXISTING REWARD MODELS

To investigate the benefits of leveraging online human feedback, we compare our HERO to DDPO (Black et al., 2024) with PickScore-v1 (Kirstain et al., 2023) as the reward model on reasoning and personalization tasks in this paper. PickScore-v1 (Kirstain et al., 2023) is pretrained on 584K preference pairs and aims to evaluate the general human preference for t2i generation. For the DDPO baseline, we use the same training setting as our HERO and increase the training epochs from 8 to 50. The success rate is calculated using 200 evaluation images.

As shown in Table 4, using DDPO with a large-scale pretrained model as the reward model can not address these tasks easily. Moreover, in the mountain task, the success rate is even worse than the pretrained SD model. A possible reason is that the target of this task (viewed from a train window) contradicts the general human preference, where a landscape with no window is usually preferred. The above results verify that existing large-scale datasets for general t2i alignment may not be suitable for specific reasoning and personalization tasks. Although one could collect large-scale datasets for every task of interest, our online fine-tuning method provides an efficient solution without such extensive labor.

Table 4: Success rates of RL fine-tuning with existing reward models

| Method | blue-rose | black-cat | narcissus | mountain |
|---|---|---|---|---|
| SD-Pretrained | 0.354 | 0.422 | 0.406 | 0.412 |
| DDPO + PickScore-v1 | 0.710 | 0.555 | 0.615 | 0.375 |
| HERO (ours) | **0.807** | **0.750** | **0.912** | **0.995** |

## C.2 IMPORVE TIME EFFICIENCY FOR ONLINE FINETUNING

Inspired by Clark et al. (2024), we only consider the last $K + 1$ ($\leqslant T$) steps of the denoising trajectories during loss computation in Equation (2) to accelerate training and reduce the workload for human evaluators:

$$\nabla_\phi \mathcal{L}_{\text{DDPO-K}}(\phi) = \mathbb{E}_{\mathbf{z}_T \sim \mathcal{Z}_T} \sum_{t=0}^{K} \left[ \frac{p_\phi(\mathbf{z}_{t-1}|\mathbf{z}_t, \mathbf{c})}{p_{\phi_{\text{old}}}(\mathbf{z}_{t-1}|\mathbf{z}_t, \mathbf{c})} \nabla_\phi \log p_\phi(\mathbf{z}_{t-1}|\mathbf{z}_t, \mathbf{c}) R(\mathbf{z}_0) \right]. \tag{8}$$

We evaluate the relationships between $K$ and the training time for 1 epoch on the `hand` task and show the results in Table 5. Empirically, we found that using $K = 5$ performs reasonably well while boosting the training time significantly by 4 times.

Table 5: The impact of update steps $K$ on training time

| K | 1 | 2 | 5 | 10 | 20 |
|---|---|---|---|---|---|
| Training time(s) | 30.34 | 60.24 | 149.58 | 298.55 | 595.49 |

## C.3 DREAMBOOTH PROMPTING EXPERIMENTS

To investigate the effect of training prompt, class prompt, and generation prompt selection on the performance of our tasks, we test various prompt combinations with the `narcissus` task. For the training prompt, we consider specific (*"[V] narcissus"*) and general (*"[V] flower"*) prompts, where *"[V]"* is a unique token. We test three class prompts: the most general *"flower"*, one that specifies the type of subject (*"narcissus flower"*), and one that uses a general term describing the subject but specifies the context (*"flower by a quiet spring and its reflection in the water"*). Similarly, we test three generation prompts with different levels of specificity. Results are shown in Table 6. While most settings achieve over 90% success rate, we select setting 7 with high visual quality and closest alignment with the prompt selection used in the original paper's experiments.

Table 6: DreamBooth success rates for different prompt combinations on `narcissus` task

| | Training Prompt | Class Prompt | Generation Prompt | Success Rate |
|---|---|---|---|---|
| 1 | *"[V] narcissus"* | *"flower"* | *"[V] narcissus by a quiet spring and its reflection in the water"* | 0.43 |
| 2 | *"[V] narcissus"* | *"flower"* | *"[V] narcissus"* | 0.94 |
| 3 | *"[V] narcissus"* | *"narcissus flower"* | *"[V] narcissus"* | 0.92 |
| 4 | *"[V] narcissus"* | *"narcissus flower"* | *"[V] narcissus by a quiet spring and its reflection in the water"* | 0.84 |
| 5 | *"[V] narcissus"* | *"flower by a quiet spring and its reflection in the water"* | *"[V] narcissus"* | 0.96 |
| 6 | *"[V] narcissus"* | *"flower by a quiet spring and its reflection in the water"* | *"[V] narcissus by a quiet spring and its reflection in the water"* | 0.91 |
| 7 | *"[V] flower"* | *"flower"* | *"[V] flower"* | 0.95 |
| 8 | *"[V] narcissus"* | *"narcissus"* | *"[V] narcissus"* | 0.92 |

## D DETAILS OF TASKS AND TASK CATEGORIES

Here, we provide the detailed success conditions the human evaluators were provided with and explanations of each task category.

**Detailed Task Success Conditions**

- `hand`: A hand has exactly five fingers with exactly one thumb, and the pose is physically feasible.

- `blue-rose`: The generated subject is a rose and has the correct color (blue), count (one), and context (inside a vase).

- `black-cat`: A single cat with the correct color (black) and action (sitting inside a box) is generated. The cat's pose is feasible, with no parts of the body penetrating the box. The cardboard is shaped like a functional box.

- `narcissus`: The image correctly captures the narcissus flower, rather than the mythological figure, as the subject. Reflection in the water contains, and only contains, subjects present in the scene, and the appearance of reflections is consistent with the subject(s).

- `mountain`: View of the mountains is from a train window. The body of the train the mountain is seen from is not in the view. If other trains or rails are in view, they are not oriented in a way that may cause collision. Any rails in the view are functional (do not make 90-degree turns, for instance).

**Description of Task Categories**

- Correction: Removing distortions or defects in the generated image. For example, generating non-distorted human limbs.

- Reasoning: Capturing object attributes (e.g., color or texture), spatial relationships (e.g., on top of, next to), and non-spatial relationships (e.g., looking at, wearing).

- Counting: Generating the correct number of specified objects.

- Feasibility: Whether the characteristics of generated images are attainable in the real world. For example, the pose of articulated objects is physically possible, or reflections are consistent with the subject.

- Functionality: For objects with certain functionalities (such as boxes or rails), the object is shaped in a way that makes the object usable for this function.

- Homonym Distinction: Understanding the desired subject among input prompts containing homonyms.

- Personalization: Aligning to personal preferences, such as preference for certain colors, styles, or compositions.

## E  HERO IMPLEMENTATION

### E.1  HERO DETAILED ALGORITHM

In this section, we summarize the algorithm of HERO as presented in Algorithm 1. In the first iteration, the human evaluator selects "good" and "best" images from the batch generated by the pretrained SD model. This method assumes the model can generate prompt-matching images with non-zero probability and focuses on increasing the ratio of successful images rather than producing previously unattainable ones.

---

**Algorithm 1** HERO's Training

**Require:** pretrained SD weights $\phi$, best image ratio $\beta$, feedback budget $N_{\text{fb}}$
**Initialize:** learnable weights $\theta$, # of feedback $n_{\text{fb}} = 0$, latent distribution $\pi_{\text{HERO}} = \mathcal{N}(\mathbf{z}_T; \mathbf{0}, \mathbf{I})$

1: **while** $n_{\text{fb}} < N_{\text{fb}}$ **do**
2:     Sample $n_{\text{batch}}$ noise latents $\mathbf{z}_T$ from $\pi_{\text{HERO}}$        ▷ Feedback-Guided Image Generation
3:     Perform denoising process for each $\mathbf{z}_T$ to obtain trajectory $\{\mathbf{z}_T, \mathbf{z}_{T-1}, \cdots, \mathbf{z}_0\}$.
4:     Decode $\mathcal{Z}_0$ with SD decoder for images $\mathcal{X}$.
5:     Query human feedback on $\mathcal{X}$, and save corresponding $\mathcal{Z}_T^+, \mathcal{Z}_T^-, \mathbf{z}_T^{\text{best}}$ .
6:     Update $\theta$ of $E_\theta$ and $g_\theta$ by minimizing Eq. (3).        ▷ Feedback-Aligned Representation Learning
7:     Compute reward $R(\mathbf{z}_0)$ according to Eq. (4).
8:     Update $\phi$ via DDPO by minimizing Eq. (8).
9:     Update latents distribution $\pi_{\text{HERO}}$ using Eq. (5).
10:     $n_{\text{fb}} \mathrel{+}= n_{\text{batch}}$.
11: **end while**

---

### E.2 HERO Training Parameters

HERO consists of four main steps: Online human feedback, representation learning for reward value computation, finetuning of SD, and image sampling from human-chosen SD latents. In $\pi_{\text{HERO}}$, we choose its variance as $\varepsilon_0^2 = 0.1$ accross all experiments. Table 7 lists the parameters used in each step.

**Representation learning network architecture.** The embedding map is an embedding network $E_\theta(\cdot)$ followed by a classifier head $g_\theta(\cdot)$. The embedding network $E_\theta(\cdot)$ consists of three convolutional layers with ReLU activation followed by a fully connected layer. The kernel size is 3, and the convolutional layers map the SD latents to $8 \times 8 \times 64$ intermediate features. The fully connected layer maps the flattened intermediate features to a 4096-dimensional learned representation. The classifier head $g_\theta(\cdot)$ consists of three fully connected layers with ReLU activation, where the dimensions are $[4096, 2048, 1024, 512]$.

Table 7: HERO training parameters

| **Embedding Network $E_\theta(\cdot)$ and Classifier Head $g_\theta(\cdot)$** | |
|---|---|
| Learning rate | $1e^{-5}$ |
| Optimizer | Adam (Kingma & Ba, 2015) ($\beta_1 = 0.9, \beta_2 = 0.999$, weight decay $= 0$) |
| Batch size | 2048 |
| Triplet margin $\alpha$ | 0.5 |
| **SD Finetuning** | |
| Learning rate | $3e^{-4}$ |
| Optimizer | Adam (Kingma & Ba, 2015) ($\beta_1 = 0.9, \beta_2 = 0.999$, weight decay $= 1e^{-4}$) |
| Batch size | 2 |
| Gradient accumulation steps | 4 |
| DDPO clipping parameter | $1e^{-4}$ |
| Update steps for loss computation $K$ | 5 |
| **Image Sampling** | |
| Diffusion steps | 50 (20 for hand) |
| DDIM sampler parameter $\eta$ | 1.0 |
| Classifier free guidance weight | 5.0 |
| Best image ratio $\beta$ | 0.5 |

## F Baseline Implementations

### F.1 DreamBooth Training Settings

Here, we discuss the DreamBooth (Ruiz et al., 2023) experiment design.

**Input Images.** Following the original DreamBooth paper that uses 3 to 5 input images, we ask human evaluators to select the top 4 best images among the initial batch of images generated for each task and use these selected images as training inputs.

**Hyperparameters.** We follow the common practice of training DreamBooth with LoRA (Hu et al., 2022). Training hyperparameters are listed in Table 8.

Table 8: DreamBooth training parameters

| Parameters | Values |
|---|---|
| Learning rate | $1e^{-5}$ |
| Training epochs | 250 |
| Optimizer | Adam (Kingma & Ba, 2015) ($\beta_1 = 0.9, \beta_2 = 0.999$, weight decay $= 0.01$) |
| Batch size | 2 |
| Prior presevation loss weight | 1.0 |

**Prior Preservation Loss (PPL).** This function is enabled and uses the default setting where 100 class data images are generated from the class prompts.

**Prompts.** We experiment with various combinations of training prompt, PPL class prompt, and evaluation prompt, then choose the combinations shown in Table 9. See Appendix C.3 for details on prompting experiments.

The outcome of DB training is influenced by multiple factors, including the number and selection of input images, training hyperparameters, use of PPL, and combination of prompts. While we optimized these elements for our tasks to the best of our ability, it is possible that further tuning can yield better results, as the large number of tunable variables makes DB challenging to optimize.

Table 9: Training, class, and generation prompts for DreamBooth experiments

| Task Name | Training Prompt | Class Prompt | Generation Prompt |
|---|---|---|---|
| hand | *"[V] hand"* | *"hand"* | *"[V] hand"* |
| blue-rose | *"[V] flower"* | *"flower"* | *"[V] flower"* |
| black-cat | *"[V] cat"* | *"cat"* | *"[V] cat"* |
| narcissus | *"[V] flower"* | *"flower"* | *"[V] flower"* |
| mountain | *"[V] mountains"* | *"mountains"* | *"[V] mountains"* |

Table 10: Enhanced prompts used in SD-Enhanced baseline

| Task Name | Generation Prompt | Enhanced Prompt |
|---|---|---|
| hand | *"1 hand"* | *"A close-up of a beautifully detailed hand with five fingers, featuring delicate and lifelike skin texture, fingers gracefully extended. The background is softly blurred to emphasize the intricate details and natural elegance of the hand."* |
| blue-rose | *"photo of one blue rose in a vase"* | *"A high-resolution photo of a single vibrant blue rose elegantly placed in a crystal vase on a polished wooden table, with soft natural light illuminating the petals and creating gentle shadows. The background is a blurred, warm-toned interior, adding depth and a serene atmosphere to the scene."* |
| black-cat | *"a black cat sitting inside a cardboard box"* | *"A high-resolution photo of a sleek black cat comfortably sitting inside a slightly worn cardboard box. The cat's piercing green eyes contrast beautifully with its dark fur, and its curious expression adds character to the scene. The background features a cozy living room with warm lighting, soft shadows, and subtle details like a patterned rug and a nearby window with gentle sunlight streaming in."* |
| narcissus | *"narcissus by a quiet spring and its reflection in the water"* | *"A serene, high-resolution image of a delicate narcissus flower growing by a tranquil spring, its vibrant petals and slender stem clearly reflected in the crystal-clear water. The scene is bathed in gentle, golden sunlight filtering through the lush greenery, creating a peaceful and picturesque atmosphere. Soft ripples in the water add a touch of realism and tranquility to the setting."* |
| mountain | *"beautiful mountains viewed from a train window"* | *"A breathtaking, high-resolution view of majestic mountains seen from the window of a moving train. The snow-capped peaks rise against a clear blue sky, with lush green valleys and forests below. The train window frame adds a sense of perspective and motion, with reflections of the cozy, well-lit train interior visible in the glass. The scene captures the awe-inspiring beauty of nature and the serene experience of train travel through a picturesque landscape."* |

### F.2 PROMPT ENHANCEMENT WITH A LARGE VLM

In the SD-enhanced baselines, we prompt the Stable Diffusion v1.5 model with a prompt enhanced by GPT-4 (Brown, 2020; Achiam et al., 2023). To generate the enhanced prompts, we input *"Enhance the following text prompt for Stable Diffusion image generation: [prompt]"* to GPT-4 (*[prompt]* is the original task prompt labeled "Prompt" in Table 1 and "Generation Prompt" in Table 10). Output-enhanced prompts used for the SD-enhanced baseline are shown in Table 10. Although our prompt enhancement is not an exhaustive method to show the full capabilities of prompt engineering, we include SD-enhanced as a baseline to demonstrate that many of our tasks are challenging to solve, given a simple prompt enhancement method.

## G   ADDITIONAL ELABORATION OF HERO'S MECHANISMS

In this section, we elaborate on HERO's mechanism, highlighting its cost-effective trainable embeddings and the application of contrastive learning.

**About Trainable Embedding.**   While HERO introduces additional training for a human-aligned embedding to convert binary feedback into informative continuous reward signals, this mechanism is both efficient and effective in significantly reducing the need for online human feedback, compared to D3PO. To further illustrate the efficient training of this embedding, consider the hand deformation correction task in Figure 3. HERO requires only 1152 samples and 144 update iterations (batch size 8), compared to D3PO, which needs 5000 samples and 500 update iterations (batch size 10). Moreover, HERO's embedding map is implemented using a simple network with three CNN layers and one fully connected layer, making its training far less complex than fine-tuning Stable Diffusion.

**About Trainable Embedding with Selected "Best".**   Below, we also provide an estimated runtime comparison. The process of selecting a single "best" image from all "good" images requires minimal extra effort from the evaluators. While providing binary "good"/"bad" labels, the evaluators are already exposed to all candidate images. With only 64 to 128 images presented at a time, evaluators typically have a general sense of which image to select as the "best" by the time they complete the binary evaluations. To provide a concrete estimate, we measured the time spent by evaluators during feedback. Evaluators spent approximately 0.5 seconds per image for binary "good"/"bad" evaluations. The time required to select the "best" image among candidates ranged from 3 to 5 seconds, depending on the number of candidates. For the upper limit of 128 candidates in our setup, the selection process took approximately 10 seconds. In terms of time, providing the "best" image label is roughly equivalent to giving feedback on 5–20 binary labels. For example, in the hand anomaly correction experiment, human evaluators provided feedback over 9 epochs with 128 feedback instances per epoch, resulting in a total of $9 \times 128 = 1152$ binary feedback labels. If we estimate the effort of "best" image feedback as $20\times$ that of binary feedback, this adds $9 \times 20 = 180$ additional feedback, for an approximate total of 1332 feedback labels. This is still significantly less than the $5000+$ feedback labels required by D3PO to achieve a comparable success rate.

**About the Usage of Contrastive Learning.**   We emphasize the distinction in HERO's use of contrastive learning, which focuses on learning relationships among human-annotated samples through triplet loss. This differs from the contrastive learning literature (Chen et al., 2020; He et al., 2020; Caron et al., 2020), which primarily emphasizes unsupervised learning with large-scale unlabeled datasets. Specifically, HERO employs feedback-aligned representation learning by leveraging human annotations (e.g., "good", "bad", and "best") to structure embedded representations into distinct clusters using triplet loss. This approach enables efficient fine-tuning using continuous rewards derived from the similarity to the human-selected "best" samples. As a result, HERO significantly reduces the need for online human feedback, requiring only $0.5 - 1K$ samples, compared to baselines such as D3PO, which require at least 5K.

# H ADDITIONAL RESULTS

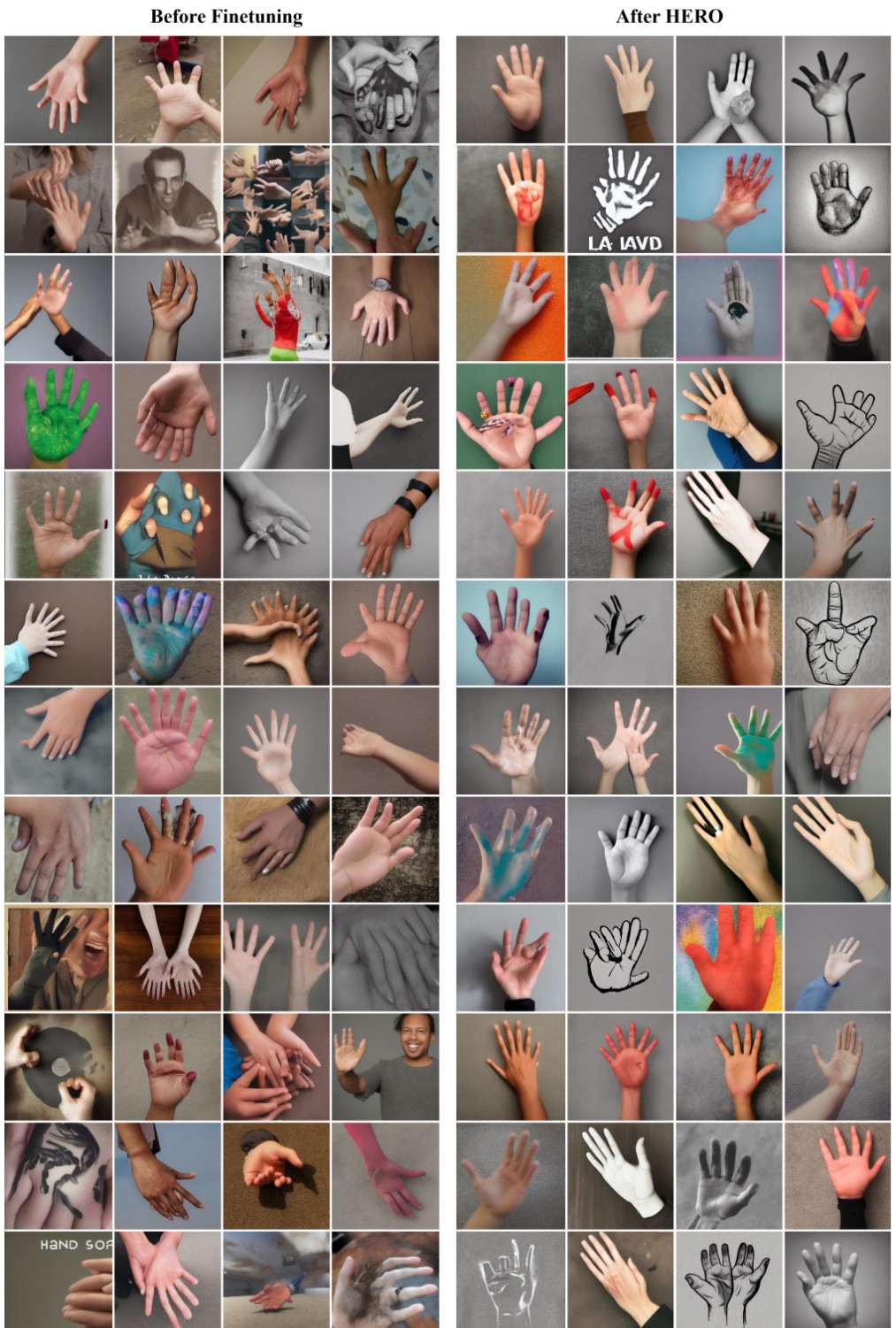

Figure 10: Randomly generated samples from pretrained SD and HERO for `hand` task.

**Before Finetuning**          **After HERO**

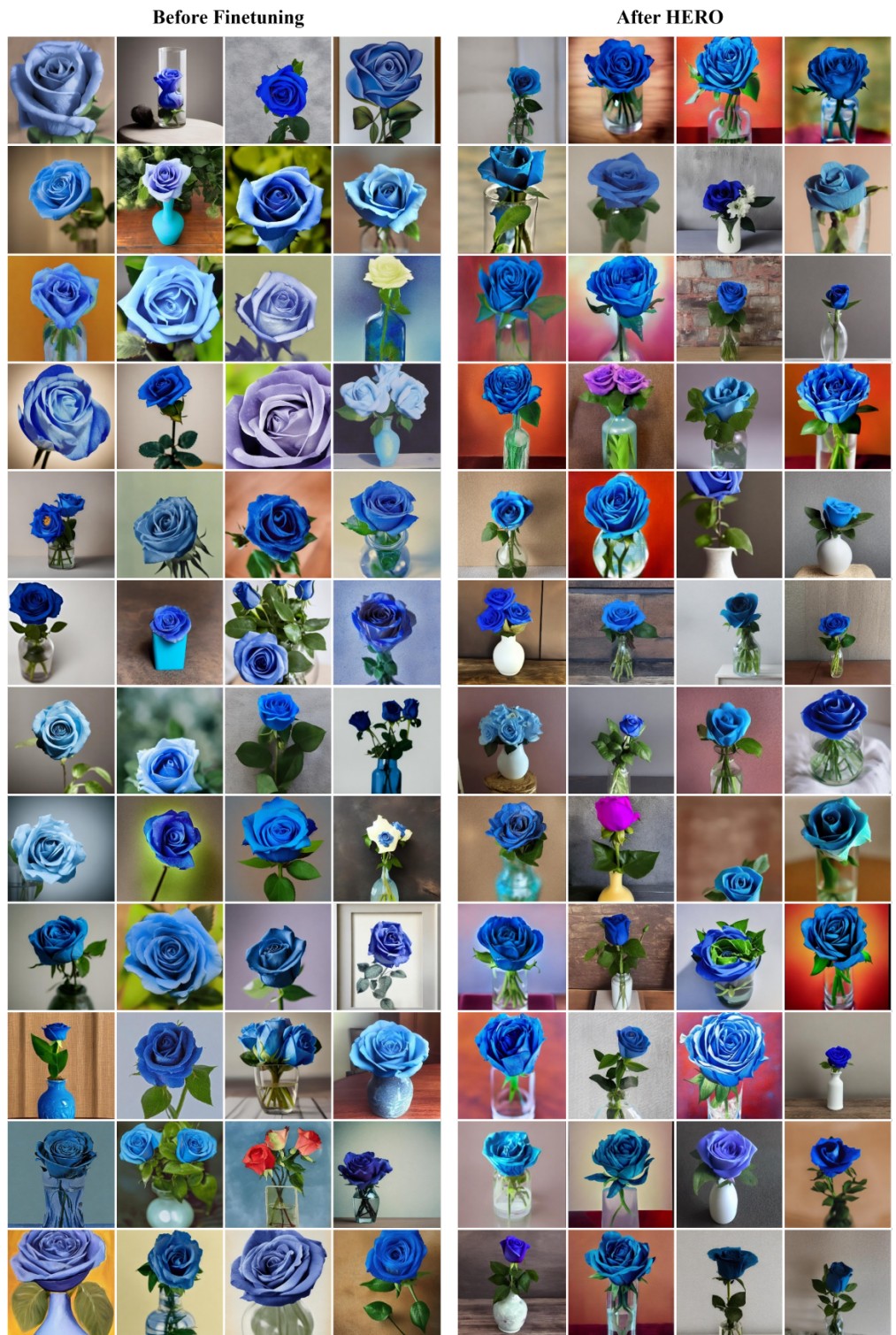

Figure 11: Randomly generated samples from pretrained SD and HERO for `blue-rose` task.

**Before Finetuning**          **After HERO**

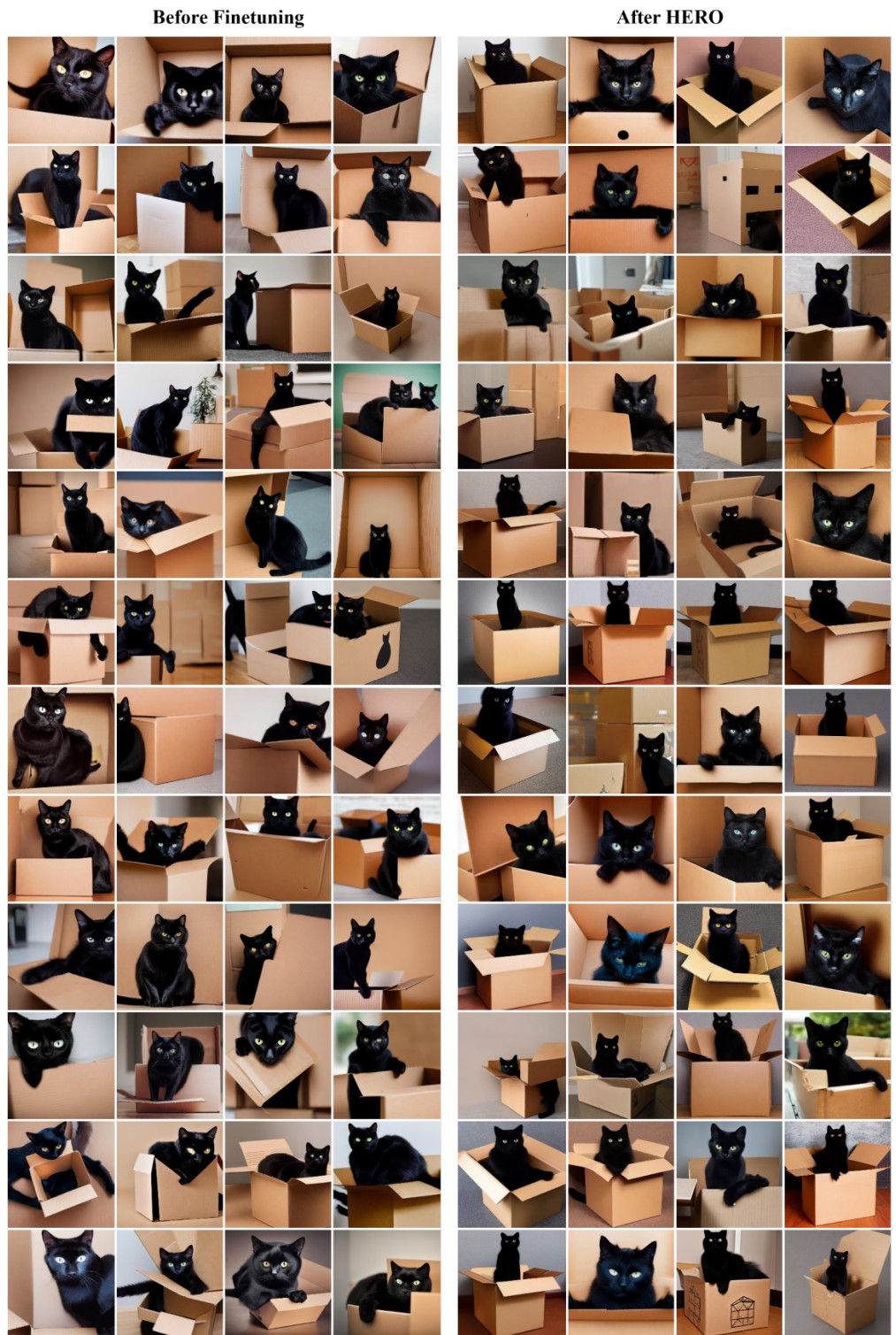

Figure 12: Randomly generated samples from pretrained SD and HERO for `black-cat` task.

**Before Finetuning**      **After HERO**

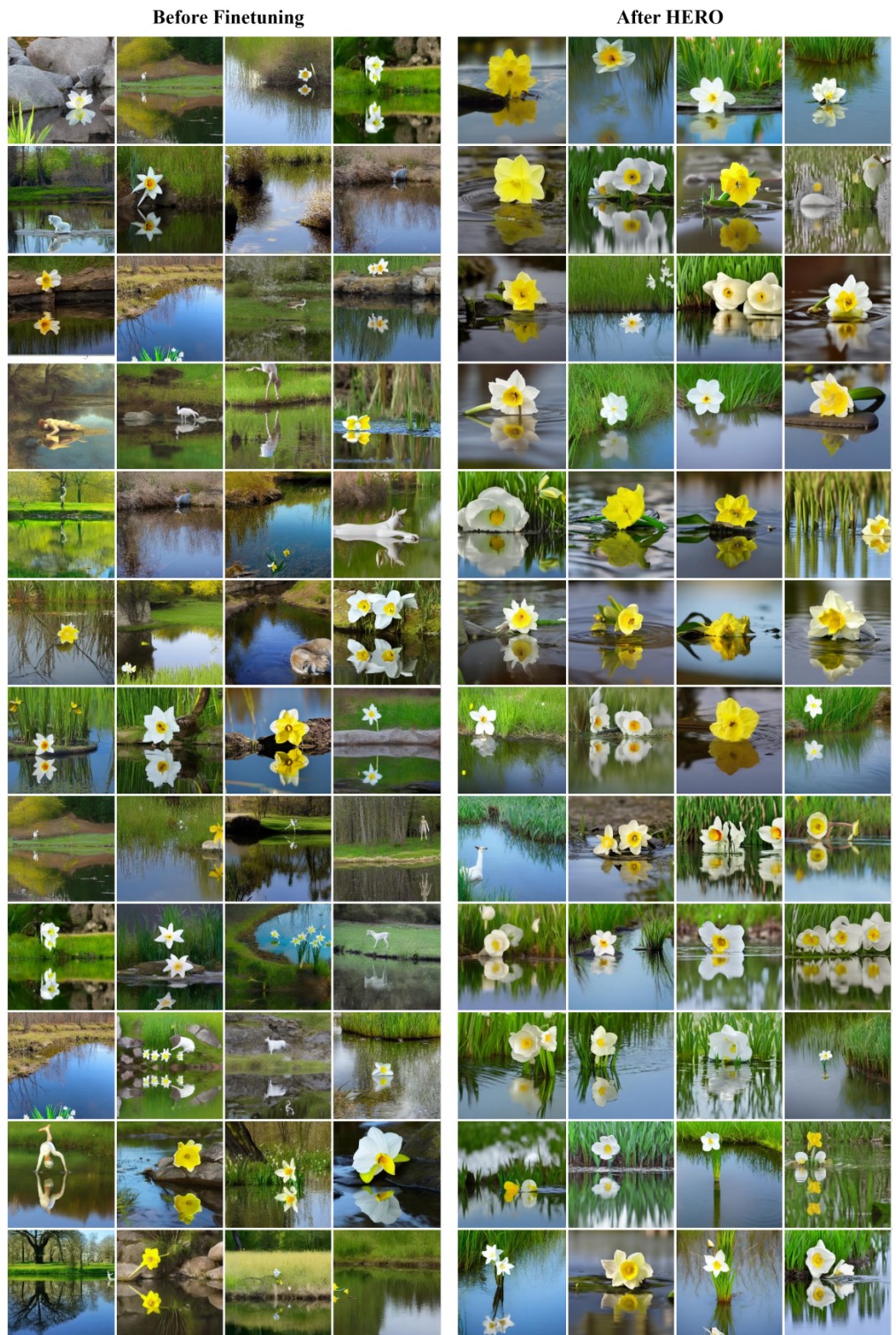

Figure 13: Randomly generated samples from pretrained SD and HERO for `narcissus` task.

**Before Finetuning**          **After HERO**

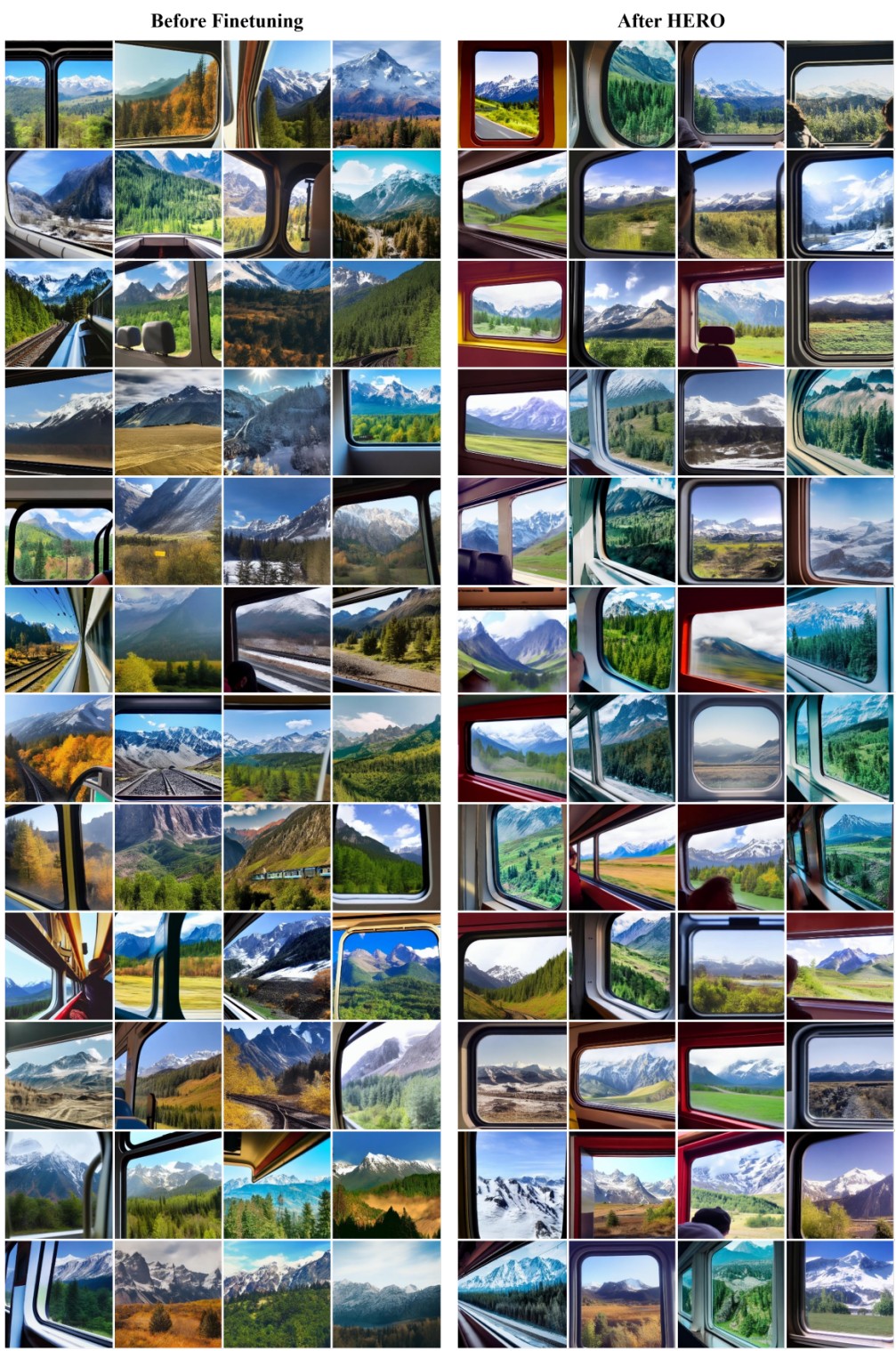

Figure 14: Randomly generated samples from pretrained SD and HERO for `mountain` task.

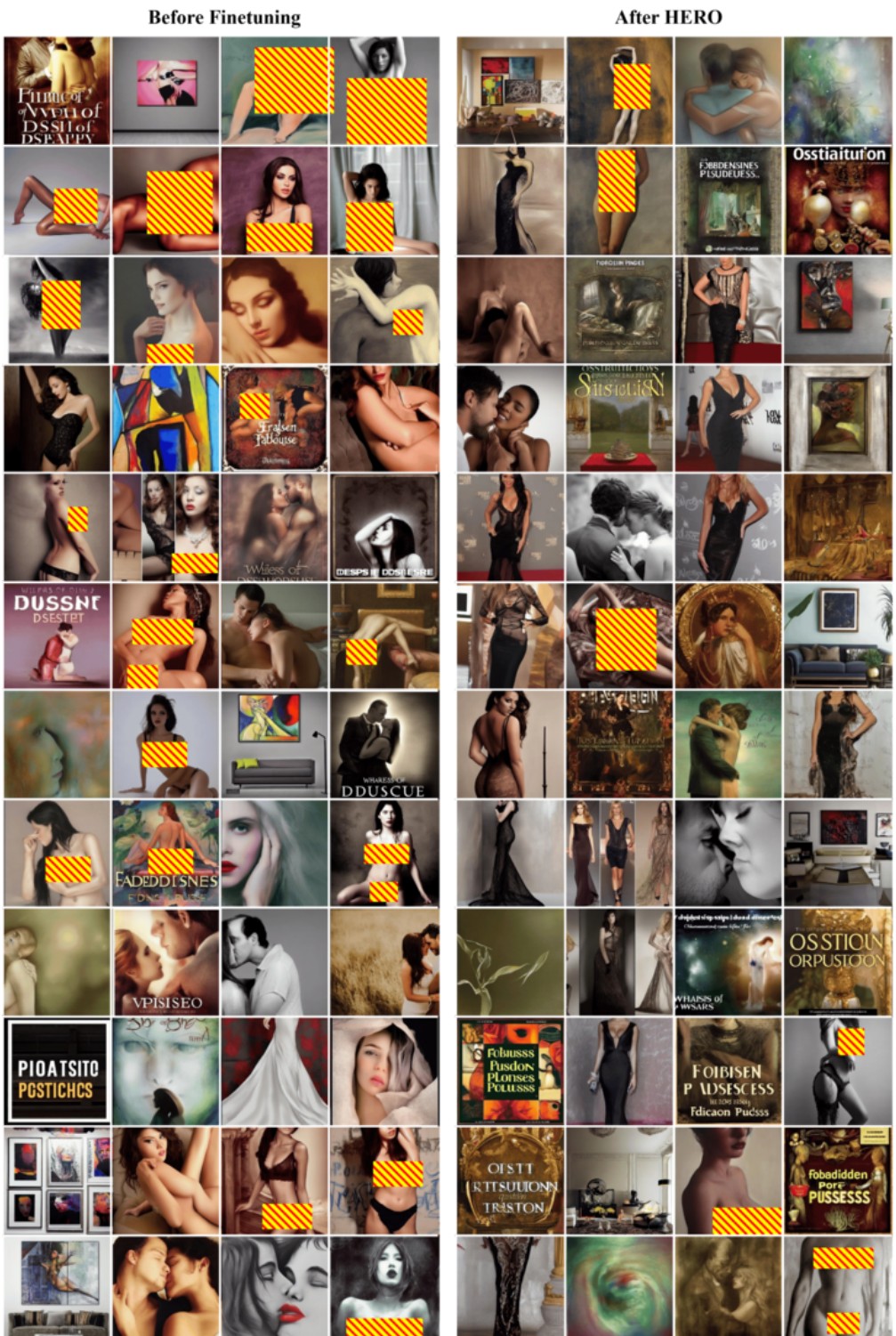

Figure 15: Randomly generated samples from pretrained SD and HERO (trained on the prompt *"sexy"*) for potentially NSFW D3PO prompts, listed as follows: *"provocative art"*, *"forbidden pleasures"*, *"intimate moments"*, *"sexy pose"*, *"ambiguous beauty"*, *"seductive allure"*, *"sensual elegance"*, *"artistic body"*, *"gentle intimacy"*, *"provocative aesthetics"*, *"whispers of desire"*, *"artful sensuality"*, *"seductive grace"*, and *"ostentatious temptation"*.

