# OpenReview forum: "HERO: Human-Feedback Efficient Reinforcement Learning for Online Diffusion Model Finetuning"
_ICLR.cc/2025/Conference — ICLR 2025 Poster_

### Official Review · Reviewer_TZdg · 2024-11-01

**Soundness:** 3
**Presentation:** 3
**Contribution:** 2
**Rating:** 6
**Confidence:** 3

**Summary:**

In this paper, the authors introduce HERO (Human-feedback Efficient Reinforcement learning for Online diffusion), a novel method for enhancing text-to-image (T2I) models using limited human feedback. The approach leverages human evaluation to refine image quality: in the data collection phase, a human annotator labels a batch of generated images with positive and negative feedback, selecting the single best image among them. HERO employs a triplet loss function to train a visual encoder by mapping embeddings based on these annotations. A reward signal, derived from the cosine similarity between the learned representation of an input image and the selected best image, guides the model optimization and image generation process. The model utilizes Proximal Policy Optimization (PPO) to apply Low-Rank Adaptation (LoRA) to a stable diffusion model. Experimental results demonstrate that HERO outperforms baseline approaches, achieving higher success rates in generating preferred images.

[Update]: Based on the additional observations across multiple metrics, revise Soundness from 2 to 3 and final review from 5 to 6.

**Strengths:**

This paper is a clear and structured presentation, which makes it easy to understand the proposed methodology and its underlying concepts.
The experiments cover a diverse set of four T2I tasks and transferability and validate the effectiveness of the proposed method.

**Weaknesses:**

1. A primary concern with the paper is that the T2I model's performance is only assessed through task success rates. Important factors such as image diversity and aesthetic quality are not quantitatively evaluated, which are crucial metrics and should be included, as seen in the baseline D3PO [1].

2. Additionally, the proposed method requires extra human labeling to identify the best image in each batch, which introduces additional information. This requirement makes direct comparison with other baselines less equitable, as they may not rely on such intensive human input.

Given these concerns, I would currently not recommend acceptance of this paper.

**Reference**

[1] Yang, Kai, et al. "Using human feedback to fine-tune diffusion models without any reward model." Proceedings of the IEEE/CVF Conference on Computer Vision and Pattern Recognition. 2024.

**Questions:**

Listed in the weakness section.

---

> ### Author Response · Authors · 2024-11-25
> **Response to Reviewer TZdg (Part 1/3)**
>
> We sincerely thank the reviewer for the thorough and constructive comments. Please find the response to your questions below.
>
> > **W1** Quantitative Evaluation Beyond Task Success Rates
>
> We extend our results by incorporating additional evaluation metrics to assess aesthetic quality, diversity, and text-to-image (T2I) alignment. We have included the additional results in the revised manuscript’s Appendix F and supplementary material to extend this discussion and provide a [link to the additional section here](https://drive.google.com/file/d/1aD0PcETlj2QY1DKPVbnK3SV42rMi3IYh/view?usp=drive_link) for convenience.
> For reference, we also include the success rate reported in our original manuscript for easier cross-reference.
> |     Success Rate          | `blackcat` | `bluerose` | `narcissus` | `mountain` | `hand` |
> | --------      | --------   | ---        | ---         | ---        | ------ |
> |SD-Pretrained  |0.4219    |0.3542    |0.4063    |0.4115    |0.1191
> |SD-Enhanced    |0.3646    |0.4792    |0.276     |0.9375    |0.0752
> |DreamBooth     |0.4531    |0.4792    |0.8542    |0.9219    |0.2803
> |HERO (Ours)    |**0.7500**|**0.8073**|**0.9115**|**0.9948**|**0.3418**
>
>
> **Additional Metrics**
>
> * Aesthetic Quality:
> We report ImageReward [1] scores, which demonstrate stronger perceptual alignment with human judgment compared to traditional metrics. Higher scores reflect better aesthetic quality.
>
> * Diversity:
> Following Section 4.3.3 of [2], we compute “In-Batch Diversity”, defined as the complement of the average similarity of CLIP image embeddings [3] between pairs of images in a generated batch. Specifically, for a batch of $N$ generated images ${I_1, I_2, \ldots, I_N}$, and the cosine similarity $\text{CLIPSim}(I_i, I_j)$ of their embeddings in the CLIP feature space, the in-batch diversity is calculated as: $D_{\text{batch}} = 1 - \frac{2}{N(N-1)} \sum_{1 \leq i < j \leq N} \text{CLIPSim}(I_i, I_j)$, where $1 - \text{CLIPSim}(I_i, I_j)$ represents the dissimilarity between two images. A higher $D_{\text{batch}}$ signifies greater diversity.
>
> * Text-to-Image Alignment:
> CLIP Score [3] evaluates the similarity between text and image embeddings, while BLIP Score [4] assesses the probability of text-to-image matching. Together, these metrics provide a quantitative measure of how well the generated images align with the given prompts. Higher scores on both metrics indicate better alignment between the generated images and the prompts.
>
> **Results:**
>
> * Aesthetic Quality:
>
> |    ImageReward           | `blackcat` | `bluerose` | `narcissus` | `mountain` | `hand` |
> | --------      | --------   | --------   | --------    | --------   | -------- |
> |SD-Pretrained | 0.4797  | 0.4463 | 0.4463 | 0.4393 | -1.3480 |
> |SD-Enhanced   | -0.4135 | -0.5911 | 0.6760 | 0.6959 | -1.4856 |
> |DreamBooth    | 0.3053  | **0.6931**  | 0.8656 | **0.9997** | -1.1078 |
> |HERO (Ours)   | **0.6394**  | -0.0031 | **1.5023** | 0.7312 | **-1.0695** |
>
>
> Although human evaluators prioritized task success based on the criteria in Appendix C over aesthetic quality and were not instructed to consider aesthetics, HERO demonstrates comparable aesthetic performance to the baselines, surpassing them in 3 out of 5 tasks.
>
> * Diversity:
>
> |    In-Batch Diversity          | `blackcat` | `bluerose` | `narcissus` | `mountain` | `hand` |
> | --------      | --------   | --------   | --------    | --------   | -------- |
> |SD-Pretrained | 0.0324 | **0.1003** | **0.0589** | **0.0941** | **0.1211** |
> |SD-Enhanced |**0.0541**| 0.0676 | 0.0499 | 0.0679 | 0.0913 |
> |DreamBooth | 0.0385 | 0.0413 | 0.0349 | 0.0567 | 0.0505 |
> |HERO (Ours) | 0.0338 | 0.0444 | 0.0537 | 0.0615 | 0.0907 |
>
> Although HERO shows a slight reduction in diversity compared to the pre-finetuned Stable Diffusion model, it generally outperforms the DreamBooth-finetuned model, except in the black-cat example (In-Batch Diversity) and mountain example (Normalized CLIP diversity). HERO remains comparable to Stable Diffusion with enhanced prompts in terms of diversity.

---

> ### Author Response · Authors · 2024-11-25
> **Response to Reviewer TZdg (Part 2/3)**
>
> * T2I Alignment:
>
> |    CLIP Score           | `blackcat` | `bluerose` | `narcissus` | `mountain` | `hand` |
> | --------      | --------   | ---        | ---         | ---        | ------ |
> |SD-Pretrained|33.7316    |29.6792    |34.4758    |27.3152    |27.3921 |
> |SD-Enhanced  |32.0388    |30.2971    |35.2811    |28.7436    |24.8990 |
> |DreamBooth   |32.9443    |31.6535    |34.4639    |**29.9026**|**27.9488** |
> |HERO (Ours)  |**33.8090**|**31.7805**|**35.5916**|28.7977    |27.7958 |
>
>
> |   BLIP Score   | `blackcat` | `bluerose` | `narcissus` | `mountain` | `hand` |
> | --------      | --------   | ---        | ---         | ---        | ------ |
> |SD-Pretrained | 0.9596 | 0.6039 | **0.9432** | 0.8037 | 0.5692 |
> SD-Enhanced | 0.6821 | 0.6834 | 0.8691 | 0.9568 | 0.2827 |
> DreamBooth | 0.8885 | 0.9139 | 0.7963 |0.9445 | 0.7087 |
> HERO (Ours) | **0.9876** | **0.9492** | 0.9407 | **0.9685** | **0.7481** |
>
>
> HERO’s finetuned model generally produces images that are more aligned with the given prompts.
>
> **Summary:**
>
> HERO demonstrates a strong balance across all metrics, excelling in task success rate while maintaining solid performance in aesthetic quality, diversity, and text-to-image alignment.
>
> [1] Xu, Jiazheng, et al. "Imagereward: Learning and evaluating human preferences for text-to-image generation." NeurIPS. 2024.
>
> [2] Von Rütte, Dimitri, et al. "Fabric: Personalizing diffusion models with iterative feedback." arXiv preprint. 2023.
>
> [3] Radford, Alec, et al. "Learning transferable visual models from natural language supervision." ICML. 2021.
>
> [4] Li, Junnan, et al. "Blip: Bootstrapping language-image pre-training for unified vision-language understanding and generation." ICML. 2022.

---

> ### Author Response · Authors · 2024-11-25
> **Response to Reviewer TZdg (Part 3/3)**
>
> > **W2** the proposed method requires extra human labeling to identify the best image in each batch, which introduces additional information. This requirement makes direct comparison with other baselines less equitable, as they may not rely on such intensive human input.
>
> **[HERO’s Key Aspect of Asking “Best”]**
>
> Asking human evaluators to select the “best” image provides additional information that is crucial for making HERO a human-feedback-efficient framework. Indeed, we consider this additional label to be one of our key contributions, as it is closely tied to:
> 1. **Feedback-Aligned Representation Learning**: Training an embedding map via contrastive learning to convert feedback into continuous representations.
> 2. **Feedback-Guided Image Generation**: Generating images from a Gaussian mixture centered around the recorded noises of “best” and “good” images (with a theoretical explanation provided in Appendix A).
> The effectiveness of the above designs is demonstrated through the ablation studies presented in Table 3 and Figure 5 in Section 5.3.
>
> **[Rough Runtime Comparison]**
>
> The process of selecting a single “best” image from all “good” images requires only minimal extra effort from the evaluators. While providing binary “good”/“bad” labels, the evaluators are already exposed to all candidate images. With only 64 to 128 images presented at a time, evaluators typically have a general sense of which image to select as the “best” by the time they complete the binary evaluations.
> To provide a concrete estimate, we measured the time spent by evaluators during feedback. Evaluators spent approximately *0.5 seconds* per image for binary “good”/“bad” evaluations. The time required to select the “best” image among candidates ranged from **3 to 5 seconds**, depending on the number of candidates. For the upper limit of 128 candidates in our setup, the selection process took approximately *10 seconds*. In terms of time, providing the “best” image label is roughly equivalent to giving feedback on *5–20 binary labels*. For example, in the hand anomaly correction experiment, human evaluators provided feedback over 9 epochs with 128 feedback per epoch, resulting in a total of *9 x 128 = 1152 binary feedback labels*. If we estimate the effort of “best” image feedback as 20x that of binary feedback, this adds *9 × 20 = 180* additional feedback, for an approximate total of *1332 feedback labels*. This is still significantly less than the *5000+ feedback labels* required by D3PO to achieve a comparable success rate.
>
> **[Conclusion]**
>
> While time is not the sole factor determining the intensity of human feedback, we emphasize that the “best” image label, which can be obtained without significant additional effort, provides valuable information that contributes to feedback efficiency. HERO’s design to effectively leverage this additional label is a core contribution to our work. Notably, we do not see a straightforward way to incorporate this feature into baseline methods without significantly altering their designs, which would make them no longer the original baselines. This further underscores the uniqueness of our approach.

---

> > ### Comment · Reviewer_TZdg · 2024-11-28
> >
> > I appreciate the additional observations and the clarification provided by the authors regarding the collection of best image feedback. I strongly encourage the authors to include this discussion in the revised version of the paper and to emphasize the different experimental settings employed in this work. Additionally, I would find it valuable if the authors could include reward curves and demonstrate how the newly provided metrics evolve with the number of samples. This would offer greater clarity and insight into the training process for readers.

---

> > > ### Author Response · Authors · 2024-11-29
> > > **Response to Reviewer TZdg's Official Comment (Part 1/2)**
> > >
> > > We sincerely appreciate the reviewer’s constructive comments. Although the paper revision period has concluded, we have incorporated this discussion into the revised manuscript. The reviewer may access it via the anonymous link to the file `ICLR_2025_HERO_DM.pdf` here: [Supplementary Material](https://drive.google.com/drive/folders/122EQ64wLiXhoody7YruFVBFyT0sW5rex).
> > >
> > > Below, we provide supplementary results in the form of tables illustrating different metrics versus training epochs (number of samples). For better visualization, the corresponding curves are also included in the file `AppendixFandG.pdf`, accessible at the same anonymous URL: [Supplementary Material](https://drive.google.com/drive/folders/122EQ64wLiXhoody7YruFVBFyT0sW5rex).
> > >
> > > **[Results of Tables]**
> > > - `blackcat`
> > >
> > > | Metric                         | Epoch 0 (Pretrained SD) | Epoch 1  | Epoch 2  | Epoch 3  | Epoch 4  | Epoch 5  | Epoch 6  | Epoch 7  | Epoch 8  |
> > > |---------------------------------|-----------|-----------|-----------|-----------|-----------|-----------|-----------|-----------|-----------|
> > > | (Aesthetic) ImageReward        | 0.4797   | 0.6194   | 0.6238   | 0.6039   | 0.6023   | 0.6375   | 0.6136   | 0.6173   | 0.6394   |
> > > | (Diversity) In-Batch Diversity | 0.0324   | 0.0309   | 0.0292   | 0.0287   | 0.0306   | 0.0319   | 0.0342   | 0.0366   | 0.0338   |
> > > | (Alignment) CLIP Score         | 33.7316  | 33.6880  | 33.6348  | 33.8242  | 33.6523  | 33.5913  | 33.5767  | 33.6376  | 33.8090  |
> > > | (Alignment) BLIP Score         | 0.9596   | 0.9771   | 0.9880   | 0.9878   | 0.9887   | 0.9876   | 0.9863   | 0.9860   | 0.9876   |
> > >
> > > - `bluerose`
> > >
> > > | Metric       | Epoch 0  (Pretrained SD)  | Epoch 1    | Epoch 2    | Epoch 3    | Epoch 4    | Epoch 5    | Epoch 6    | Epoch 7    | Epoch 8    |
> > > | ------------------------------ |-------------|-------------|-------------|-------------|-------------|-------------|-------------|-------------|-------------|
> > > | (Aesthetic) ImageReward      | 0.4463 | 0.9337 | 0.9852 | 0.8063 | 0.7960 | 1.1006 | 1.4675 | 1.5441 | 1.5023 |
> > > | (Diversity) In-Batch Diversity | 0.0589 | 0.0606 | 0.0595 | 0.0584 | 0.0521 | 0.0535 | 0.0570 | 0.0556 | 0.0537 |
> > > | (Alignment) CLIP Score  | 34.4758 | 34.9937 | 35.0022| 34.5263 | 34.1363 | 34.7620 | 35.5396 | 35.4977 | 35.5916 |
> > > | (Alignment) BLIP Score       | 0.9432 | 0.9405 | 0.9666 | 0.9556 | 0.9419 | 0.9623 | 0.9506 | 0.9618 | 0.9407 |
> > >
> > > - `narcissus`
> > >
> > > | Metric                     | Epoch 0 (Pretrained SD)  | Epoch 1   | Epoch 2   | Epoch 3   | Epoch 4   | Epoch 5   | Epoch 6   | Epoch 7   | Epoch 8   |
> > > |-----------------------------|------------|------------|------------|------------|------------|------------|------------|------------|------------|
> > > | (Aesthetic) ImageReward    | -0.6015 | -0.1204| -0.1828| 0.1357 | 0.3258 | 0.4096 | 0.2431 | 0.0346 | -0.0045|
> > > | (Diversity) In-Batch Diversity | 0.1003 | 0.0854 | 0.0749  | 0.0627 | 0.0540 | 0.0433 | 0.0471 | 0.0478 | 0.0444 |
> > > | (Alignment) CLIP Score     | 29.6792 | 31.3040| 31.3877| 32.0722 | 32.1730 | 32.1188 | 31.9487| 31.7274| 31.7805|
> > > | (Alignment) BLIP Score     | 0.6039 | 0.8115  | 0.8068 | 0.8851 | 0.9250 | 0.9280 | 0.9360 | 0.9129 | 0.9492 |
> > >
> > > - `mountain`
> > >
> > > | Metric                      | Epoch 0 (Pretrained SD) | Epoch 1  | Epoch 2  | Epoch 3  | Epoch 4  | Epoch 5  | Epoch 6  | Epoch 7  | Epoch 8  |
> > > |------------------------------|-----------|-----------|-----------|-----------|-----------|-----------|-----------|-----------|-----------|
> > > | (Aesthetic) ImageReward     | 0.4393   | 0.6075   | 0.6778   | 0.7638   | 0.7520   | 0.7615   | 0.6720   | 0.7072   | 0.7312   |
> > > | (Diversity) In-Batch Diversity | 0.0941   | 0.0810   | 0.0638   | 0.0681   | 0.0619   | 0.0647   | 0.0613   | 0.0605   | 0.0615   |
> > > | (Alignment) CLIP Score      | 27.3152  | 28.1703  | 29.0835  | 29.1750  | 29.1832  | 29.0779  | 28.9096  | 28.9625  | 28.7977  |
> > > | (Alignment) BLIP Score      | 0.8037   | 0.9138   | 0.9716   | 0.9679   | 0.9843   | 0.9715   | 0.9670   | 0.9718   | 0.9685   |
> > >
> > > - `hand`
> > >
> > > | Metric              | Epoch 0  (Pretrained SD)   | Epoch 1     | Epoch 2     | Epoch 3     | Epoch 4     | Epoch 5     | Epoch 6     | Epoch 7     | Epoch 8     | Epoch 9     |
> > > |-----------------------|--------------|--------------|--------------|--------------|--------------|--------------|--------------|--------------|--------------|--------------|
> > > | (Aesthetic) Image Reward        | -1.3380 | -1.2949 | -1.2030 | -1.1683 | -1.1656 | -1.1107 | -1.0637  | -1.0809 | -1.0671 | -1.0351 |
> > > | (Diversity) In-Batch Diversity  | 0.1209  | 0.1156    | 0.1070  | 0.1095  | 0.0982  | 0.0964  | 0.0875  | 0.0880   | 0.0932   | 0.0852  |
> > > | (Alignment) CLIP Score          | 27.3371 | 27.6565 | 27.8736 | 27.9376  | 27.9022 | 28.0212   | 27.9770 | 27.9616 | 27.9155 | 28.0276 |
> > > | (Alignment) BLIP Score          | 0.5735  | 0.6104  | 0.6525   | 0.6959  | 0.7120  | 0.7316  | 0.7652  | 0.7629   | 0.7649  | 0.7786  |

---

> > > ### Author Response · Authors · 2024-11-29
> > > **Response to Reviewer TZdg's Official Comment (Part 2/2)**
> > >
> > > **[Discussions]**
> > >
> > > From the result tables (above) and the curves (linked), we generally observe the following:
> > > - **Aesthetic** (measured with ImageReward): Aesthetic quality is generally maintained throughout the fine-tuning process, demonstrating that HERO does not compromise aesthetic appeal even with increased human feedback.
> > > - **Diversity** (measured with In-Batch Diversity Score): As HERO fine-tuning progresses, the generated outputs may become more aligned with human intentions, potentially reducing diversity. This aligns with the common phenomenon where stronger guidance often leads to lower diversity. Note that HERO still generally outperforms the DreamBooth-finetuned model regarding to the diversity score.
> > > - **Alignment** (measured with CLIP and BLIP Scores): The alignment between prompts and generated images consistently improves with HERO fine-tuning. This provides implicit evidence that HERO fine-tuning effectively converges toward human intention, as reflected in the prompts.
> > >
> > > We hope these insights help clarify our findings. Please feel free to reach out if you have further questions or require additional clarifications.

---

> > > > ### Comment · Reviewer_TZdg · 2024-11-29
> > > >
> > > > I appreciate the additional observations provided by the authors. Taking these into account, I am happy to revise my score from 5 to 6. Thank you.

---

> > > > > ### Author Response · Authors · 2024-11-30
> > > > > **Thank you**
> > > > >
> > > > > We sincerely appreciate the reviewers' feedback in helping us improve this work. If there are any further questions requiring clarification, we would be happy to address them.

---

### Official Review · Reviewer_UoHW · 2024-11-01

**Soundness:** 4
**Presentation:** 4
**Contribution:** 3
**Rating:** 6
**Confidence:** 4

**Summary:**

This paper introduces HERO, a new framework for fine-tuning Stable Diffusion models using online human feedback efficiently. HERO integrates two novel components: Feedback-Aligned Representation Learning and Feedback-Guided Image Generation. These components are designed to maximize learning efficiency by converting human judgments into informative training signals, thereby reducing the reliance on large pre-trained models or extensive heuristic datasets. The model demonstrates significant improvements in online learning efficiency, requiring considerably fewer instances of human feedback compared to previous methods, while effectively enhancing image generation aligned with human preferences.

**Strengths:**

1. Efficiency in Feedback Use: HERO significantly reduces the need for human feedback instances by using them more effectively compared to previous methods, such as D3PO.
2. Direct Use of Human Judgments: By converting direct human feedback into learning signals without the need for pre-trained models, HERO simplifies the training process and potentially increases the model's responsiveness to nuanced human evaluations.
3. Improved Learning from Sparse Data: The methodology allows for effective learning even when limited data is available, which is a critical advantage in scenarios where generating or collecting extensive labeled datasets is impractical or impossible.

**Weaknesses:**

1. Algorithmic Complexity: The incorporation of sophisticated mechanisms like contrastive learning and feedback-based sampling may introduce complexity that complicates the model's implementation and optimization, potentially requiring specialized knowledge or resources to manage effectively.
2. Sensitivity to Feedback Quality: The performance of HERO heavily depends on the relevance and accuracy of the feedback provided. Inconsistent or poor-quality feedback could mislead the learning process, leading to suboptimal or biased model behavior.

**Questions:**

1. How can HERO be adapted to remain robust against noisy or contradictory feedback, which is common in real-world scenarios?
2. Could there be a hybrid approach that integrates automated feedback mechanisms with human judgments to reduce dependency on constant human input while retaining the benefits of nuanced understanding?
3. How transferable is the HERO framework across different domains or types of generative models? Can the principles applied here be adapted for use in non-visual tasks, such as text generation or music synthesis?

---

> ### Author Response · Authors · 2024-11-25
> **Response to Reviewer UoHW**
>
> We sincerely thank the reviewer for the thorough and constructive comments. Please find the response to your questions below.
>
> > **W1** Algorithmic Complexity: The incorporation of sophisticated mechanisms like contrastive learning and feedback-based sampling may introduce complexity that complicates the model's implementation and optimization, potentially requiring specialized knowledge or resources to manage effectively.
>
> HERO introduces an additional human-aligned embedding to convert the binary feedback into informative continuous reward signals. The embedding map is implemented with a simple network comprising three CNN layers and one fully connected layer, making its training process far less complex than fine-tuning Stable Diffusion. We are happy to open-source the codebase for reproducibility after acceptance.
>
> > **W2/Q1** The performance of HERO heavily depends on the relevance and accuracy of the feedback provided. Inconsistent or poor-quality feedback could mislead the learning process, leading to suboptimal or biased model behavior. How can HERO be adapted to remain robust against noisy or contradictory feedback, which is common in real-world scenarios?
>
> We agree with the reviewer that HERO's performance relies on the quality of the provided feedback, as is common with methods that learn from human input. While real human feedback inherently contains noise, we demonstrate that HERO has a certain level of robustness by showing that 3 different human evaluators can all finetune the Stable Diffusion model to a high success rate for various tasks with acceptable standard deviation in Table 2.
>
> Moreover, a key motivation behind HERO is to minimize the amount of required feedback, enabling the fine-tuning of Stable Diffusion with input from a single evaluator. When all feedback is provided by the same individual, contradictory feedback is less likely to exist, and annotation preferences tend to be more consistent than methods that rely on multiple evaluators for a specific task.
>
>
> > **Q2** Could there be a hybrid approach that integrates automated feedback mechanisms with human judgments to reduce dependency on constant human input while retaining the benefits of nuanced understanding?
>
> We thank the reviewer for bringing up this interesting idea! Combining automated AI feedback with human feedback could indeed be an effective way to further reduce the need for human input. While it is feasible to incorporate such a hybrid approach into HERO, it introduces unique challenges.
>
> **[General vs. Individual Preference]**
>
> One of the main motivations behind HERO is to fine-tune Stable Diffusion based on feedback from a single evaluator, capturing their individual preferences. In contrast, AI models are typically trained on large offline datasets annotated by multiple evaluators. Therefore, feedback from the AI model may be impractical for certain tasks of interest and could introduce inconsistencies between the general preferences of the AI evaluator and the specific preferences of the individual evaluator.
>
> **[Unbalanced amount of feedback samples]**
>
> An AI evaluator can provide numerous inexpensive feedback while human evaluators can provide few but potentially higher quality feedback. Addressing data imbalance and effectively leveraging higher-quality human feedback remains a challenge.
>
> **[Feedback format]**
>
> Aligning the format of feedback—whether binary, discrete scores (e.g., 1-10) or continuous values—between AI models and humans is not always straightforward or practical.
>
> Despite these potential challenges, exploring such a hybrid approach could be an interesting direction for future work.
>
> > **Q3** How transferable is the HERO framework across different domains or types of generative models? Can the principles applied here be adapted for use in non-visual tasks, such as text generation or music synthesis?
>
> We believe HERO is well-suited for Continuous Latent Diffusion Models, as the structured latent space provided by pre-trained autoencoders with continuous latent features aligns naturally with our novel approaches: (1) Efficient Feedback-Aligned Representation Learning in Latent Space, an online training method that captures human feedback and provides informative learning signals for fine-tuning, and (2) Feedback-Guided Image Generation, which generates images from Stable Diffusion’s refined initialization samples, enabling faster convergence to the evaluator’s intent.
>
> In this spirit, while our work focuses on text-to-image generation, we believe HERO can be extended to other domains, such as text or music, provided they are supported by sufficiently effective pre-trained continuous latent diffusion models. Exploring this extension would be an exciting direction to pursue!

---

> ### Author Response · Authors · 2024-11-29
> **A Kind Reminder for Comments**
>
> We would like to sincerely thank the reviewer for their thorough and constructive feedback. We are confident that our responses address the concerns raised, particularly the following points:
>
> - A clarification of HERO’s algorithm complexity, highlighting its simplicity and effectiveness, as well as its low-cost embedding map training.
> - An explanation of sensitivity to feedback quality. We show HERO’s robustness across different human evaluators, with an emphasis on how HERO aims to reduce the amount of feedback from any single evaluator, minimizing the impact of noisy feedback.
> - An elaboration on how automatic feedback is integrated with human feedback in HERO.
> - A discussion of HERO’s applicability to other modalities.
>
> Please let us know if the reviewer has any additional concerns or if further experimental results are needed. We are committed to addressing any remaining issues, time permitting. Once again, we thank the reviewer for their detailed feedback and the time invested in helping us improve our submission.

---

> ### Author Response · Authors · 2024-12-03
> **Kind Reminder**
>
> Dear Reviewer,
>
> The discussion phase will end in less than 9 hours.
>
> We are eager to know if we have fully addressed your concerns, as we believe we have.
> However, if you still have any questions or require further clarification, we kindly request that you leave a message before the rebuttal period ends.
> We would be more than happy to respond promptly.
>
>
> Best regards,
>
> The Authors

---

> > ### Author Response · Authors · 2024-12-04
> > **Still looking forward to the reviewer's feedback**
> >
> > Dear Reviewer,
> >
> >
> > As the deadline for reviewers to post a message has recently passed, we understand that it is no longer possible to provide an official comment. **However, we would be deeply grateful if the reviewer could kindly edit the original review to let us know whether our rebuttal and the revised paper adequately address the questions and concerns previously raised.** Your feedback would be invaluable in helping us further improve our submission.

---

### Official Review · Reviewer_Zkaj · 2024-11-04

**Soundness:** 2
**Presentation:** 4
**Contribution:** 3
**Rating:** 6
**Confidence:** 3

**Summary:**

The paper introduces HERO, a novel framework designed for fine-tuning diffusion models using human feedback, aimed at improving text-to-image (T2I) generation tasks. HERO uses feedback-aligned representation learning to create a latent representation space guided by human annotations. Human evaluators categorize generated images into “best,” “good,” or “bad,” to guide a contrastive learning process that constructs an embedding space. Triplet loss is applied to align embeddings of “best” and “good” images while distancing “bad” images, resulting in a reward signal that guides the model toward human preferences. The framework employs DDPO (Diffusion-based Policy Optimization) for updates and uses LoRA for parameter efficient fine-tuning. At inference, images are sampled from a Gaussian mixture model based on the noise latents of “good” and “best” images from previous iterations, balancing quality and diversity. Experimental results indicate that HERO achieves high success rates across various T2I tasks, demonstrating both sample efficiency and superior performance compared to other feedback-guided methods.

**Strengths:**

- **Originality**: HERO presents a unique extension of binary signal methodologies to continuous reward signals, effectively merging representational learning with reinforcement learning to enhance the alignment of generated images with human feedback.
- **Clarity**: The paper is well-written, with clearly labeled diagrams and detailed qualitative examples that illustrate the feedback process. The structured presentation of the HERO framework, including its iterative feedback mechanism, allows readers to easily grasp the method's operation. The paper also includes many qualitative examples, showcasing the benefit of the HERO pipeline
- **Performance**: Results demonstrate that HERO significantly enhances sample efficiency and alignment compared to previous methods, highlighting its practical impact in T2I generation tasks.
- **Flexibility across tasks:** The results suggest that the pipeline can be widely applied to a wide range of tasks - such as content safety improvement to reasoning-based generation.

**Weaknesses:**

1. **Limited Task Diversity and Complexity**:
    - Evaluation is conducted across only five T2I tasks, which is significantly less than comparable works like D3PO, which evaluated across 300 prompts.
    - Tasks are primarily simple single-object scenarios and do not encompass multi-object compositions or complex interactions. Expanding to more challenging tasks would improve the robustness of the findings.

2. **Insufficient Diversity and Convergence Analysis**:
    - The paper lacks a quantitative analysis of the diversity-quality trade-off, particularly missing comparisons between non-fine-tuned and feedback-guided generators.
    - There are no established metrics for evaluating mode collapse or potential overfitting to ideal seeds, which could limit the practical application of the generator.

3. **Concerns Regarding Human Feedback Methodology**:
    - Results are reported based on a limited number of human evaluators, with each evaluator responsible for different seeds. This implies a lack of inter-annotator agreements and introduces potential biases from relying on individual evaluators, which could skew the model's alignment and generalization capabilities.
    - There is also limited information on evaluation reliability measures, such as the criteria used for selection.

**Questions:**

Some questions and suggestions for the authors to consider:
1. Expand evaluation to include more diverse and complex tasks, particularly multi-object scenarios
2. Will the authors consider performing a thorough analysis of diversity with established metrics to better understand the trade-offs in quality?
3. Is there a plan to establish structured protocols for feedback collection that involve multiple evaluators to enhance reliability and reduce biases?
6. Would the authors be able to incorporate an analysis of failure cases along with strategies for mitigation in future work?

---

> ### Author Response · Authors · 2024-11-25
> **Response to Reviewer Zkaj (Part 1/5)**
>
> We sincerely thank the reviewer for the thorough and constructive comments. Please find the response to your questions below.
>
> > **W1** Limited Task Diversity and Complexity
>
> **[Numbers of Tasks vs. Prompts]**
> Thank you for your valuable feedback. First, we would like to clarify that the number of tasks and the number of prompts are two distinct concepts. D3PO uses multiple prompts within a single task—text-to-image alignment (T2I alignment), where the goal is to fine-tune the model to generate images that align with the provided prompts. Therefore, T2I alignment constitutes one task, but due to its nature, it requires multiple prompts for both training and evaluation. However, T2I alignment is not the primary focus of our work. Indeed, extensive literature [1,2,3,4] has demonstrated that this task can be effectively addressed using well-established large-scale pre-collected text-image datasets and can be performed offline using supervised learning without relying on online human feedback. Overall, D3PO itself involves only three main tasks with online human feedback: Reduce Image Distortion, Enhance Image Safety, and Prompt-Image Alignment.
>
> In contrast, we evaluate our HERO across five distinct tasks, each designed with a unique prompt, as summarized in Table 1 and Appendix C. These tasks include (1) Image Distortion Reduction (e.g., hand example), (2) Safety Enhancement (e.g., NSFW example), (3) Reasoning (e.g., blue-rose, black cat, mountain examples), (4) Counting (e.g., blue-rose and black cat examples), (5) Homonym Distinction (e.g., narcissus example), and (6) Personalization (e.g., mountain example). These tasks represent a wide range of applications, demonstrating that HERO addresses diverse real-world challenges.
>
> **[Multi-Object Compositions or Complex Interactions]**
> We would like to emphasize that the tasks outlined in Table 1 and Appendix C, as well as briefly explained above, inherently involve multi-object compositions and complex interactions. We further clarify the complexity of our prompt designs below:
> * Blue-Rose Example: The prompt "photo of one blue rose in a vase" requires generating exactly one blue rose inside a vase, involving multiple attributes (color & position) and two objects (blue rose and vase).
> * Black-Cat Example: The prompt "a black cat sitting inside a cardboard box" requires generating one black cat (counting & color) with no distortion or body parts penetrating the box (feasibility & functionality), sitting inside (reasoning) a functional and non-distorted cardboard box (functionality).
> * Narcissus Example: The prompt "narcissus by a quiet spring and its reflection in the water" is complex because "narcissus" is a homonym. The goal is to generate the narcissus flower, not the mythological figure, and ensure that the reflection only includes subjects present in the scene with consistent appearance (feasibility).
> * Mountain Example: The prompt "beautiful mountains viewed from a train window" asks for a view of mountains from inside a train (reasoning), ensuring no train body is visible, other trains or rails are not oriented to cause a collision, and any visible rails are functional (functionality). Additionally, in Figure 6, we show that models trained with two distinct personal preferences (green and snowy) generate images reflecting those preferences, highlighting the personalized nature of the task.
>
> These tasks present significant challenges, as they cannot be easily solved by prompt engineering alone, nor by previous methods (see Table 2) or well-established pretrained models (see Table 4 in the appendix) using large-scale datasets like Pick-a-Pic [1]. Therefore, we believe HERO effectively addresses these diverse and complex tasks, which involve multi-object compositions and intricate interactions.
>
> [1] Kirstain, Yuval, et al. "Pick-a-pic: An open dataset of user preferences for text-to-image generation." NeurIPS. 2023.
>
> [2] Lee, Kimin, et al. "Aligning text-to-image models using human feedback." arXiv preprint. 2023.
>
> [3] Xu, Jiazheng, et al. "Imagereward: Learning and evaluating human preferences for text-to-image generation." NeurIPS. 2024.
>
> [4] Liang, Youwei, et al. "Rich human feedback for text-to-image generation." CVPR. 2024.

---

> ### Author Response · Authors · 2024-11-25
> **Response to Reviewer Zkaj (Part 2/5)**
>
> > **W2** Insufficient Diversity and Convergence Analysis
>
> **[Diversity-Quality Trade-Off]**
>
> We extend our results by incorporating additional evaluation metrics to assess aesthetic quality, diversity, and text-to-image (T2I) alignment. We have included the additional results in the revised manuscript’s Appendix F and supplementary material to extend this discussion and provide a [link to the additional section here](https://drive.google.com/file/d/1aD0PcETlj2QY1DKPVbnK3SV42rMi3IYh/view?usp=drive_link) for convenience.
> For reference, we also include the success rate reported in our original manuscript for easier cross-reference:
>
> | Success Rate | `blackcat` | `bluerose` | `narcissus` | `mountain` | `hand` |
> | ---  | ---  | --- | --- | --- | --- |
> |SD-Pretrained  |0.4219    |0.3542    |0.4063    |0.4115    |0.1191
> |SD-Enhanced    |0.3646    |0.4792    |0.276     |0.9375    |0.0752
> |DreamBooth     |0.4531    |0.4792    |0.8542    |0.9219    |0.2803
> |HERO (Ours)    |**0.7500**|**0.8073**|**0.9115**|**0.9948**|**0.3418**
>
>
> **Additional Metrics:**
>
> - Aesthetic Quality: We report ImageReward [1] scores, which demonstrate stronger perceptual alignment with human judgment compared to traditional metrics. Higher scores reflect better aesthetic quality.
>
> - Diversity: Following Section 4.3.3 of [2], we compute “In-Batch Diversity,” defined as the complement of the average similarity of CLIP image embeddings [3] between pairs of images in a generated batch. Specifically, for a batch of $N$ generated images ${I_1, I_2, \ldots, I_N}$, and the cosine similarity $\text{CLIPSim}(I_i, I_j)$ of their embeddings in the CLIP feature space, the in-batch diversity is calculated as: $D_{\text{batch}} = 1 - \frac{2}{N(N-1)} \sum_{1 \leq i < j \leq N} \text{CLIPSim}(I_i, I_j)$, where $1 - \text{CLIPSim}(I_i, I_j)$ represents the dissimilarity between two images. A higher $D_{\text{batch}}$ signifies greater diversity.
>
> - Text-to-Image Alignment: CLIP Score [3] evaluates the similarity between text and image embeddings, while BLIP Score [4] assesses the probability of text-to-image matching. Together, these metrics provide a quantitative measure of how well the generated images align with the given prompts. Higher scores on both metrics indicate better alignment between the generated images and the prompts.
>
> **Results:**
>
> * Aesthetic Quality:
>
> | ImageReward | `blackcat` | `bluerose` | `narcissus` | `mountain` | `hand` |
> | --- | --- | --- | --- | --- | --- |
> |SD-Pretrained | 0.4797  | 0.4463 | 0.4463 | 0.4393 | -1.3480 |
> |SD-Enhanced   | -0.4135 | -0.5911 | 0.6760 | 0.6959 | -1.4856 |
> |DreamBooth    | 0.3053  | **0.6931**  | 0.8656 | **0.9997** | -1.1078 |
> |HERO (Ours)   | **0.6394**  | -0.0031 | **1.5023** | 0.7312 | **-1.0695** |
>
> Although human evaluators prioritized task success based on the criteria in Appendix C and were not instructed to consider aesthetics, HERO demonstrates comparable aesthetic performance to the baselines, surpassing them in 3 out of 5 tasks.
>
> * Diversity:
>
> | In-Batch Diversity | `blackcat` | `bluerose` | `narcissus` | `mountain` | `hand` |
> | --- | --- | --- | --- | --- | --- |
> |SD-Pretrained | 0.0324 | **0.1003** | **0.0589** | **0.0941** | **0.1211** |
> |SD-Enhanced |**0.0541**| 0.0676 | 0.0499 | 0.0679 | 0.0913 |
> |DreamBooth | 0.0385 | 0.0413 | 0.0349 | 0.0567 | 0.0505 |
> |HERO (Ours) | 0.0338 | 0.0444 | 0.0537 | 0.0615 | 0.0907 |
>
> Although HERO shows a slight reduction in diversity compared to the pre-finetuned Stable Diffusion model, it generally outperforms the DreamBooth-finetuned model, except in the black-cat example. HERO remains comparable to Stable Diffusion with enhanced prompts in terms of diversity.
>
>
> * T2I Alignment:
>
> | CLIP Score | `blackcat` | `bluerose` | `narcissus` | `mountain` | `hand` |
> | --- | --- | --- | --- | --- | --- |
> |SD-Pretrained|33.7316    |29.6792    |34.4758    |27.3152    |27.3921 |
> |SD-Enhanced  |32.0388    |30.2971    |35.2811    |28.7436    |24.8990 |
> |DreamBooth   |32.9443    |31.6535    |34.4639    |**29.9026**|**27.9488** |
> |HERO (Ours)  |**33.8090**|**31.7805**|**35.5916**|28.7977    |27.7958 |
>
> | BLIP Score | `blackcat` | `bluerose` | `narcissus` | `mountain` | `hand` |
> | --- | --- | --- | --- | --- | --- |
> |SD-Pretrained | 0.9596 | 0.6039 | **0.9432** | 0.8037 | 0.5692 |
> SD-Enhanced | 0.6821 | 0.6834 | 0.8691 | 0.9568 | 0.2827 |
> DreamBooth | 0.8885 | 0.9139 | 0.7963 |0.9445 | 0.7087 |
> HERO (Ours) | **0.9876** | **0.9492** | 0.9407 | **0.9685** | **0.7481** |
>
> HERO’s finetuned model generally produces images that are more aligned with the given prompts.

---

> ### Author Response · Authors · 2024-11-25
> **Response to Reviewer Zkaj (Part 3/5)**
>
> **Summary of the response to W2**
>
> We appreciate the reviewer’s insightful comment. Since there are no well-established metrics for directly measuring mode collapse or overfitting in guidance tasks, we have used diversity scores along with success rates as implicit indicators of potential mode collapse.
> Our results show that HERO consistently achieves higher average diversity scores and success rates across different seeds compared to the baselines (as shown in Table 2).
> While it is common for diversity scores to vary depending on the seed (as noted in [5]), we found no significant correlation between diversity scores and success rates.
> This suggests that HERO does not exhibit any preference for specific seeds to achieve better success rates nor experience mode collapse.
> Overall, HERO demonstrates a strong balance across all metrics, excelling in task success rate while maintaining solid performance in aesthetic quality, diversity, and text-to-image alignment.
>
> We thank the reviewer once again for raising these insightful questions and helping us to improve our submission.
>
> [1] Xu, Jiazheng, et al. "Imagereward: Learning and evaluating human preferences for text-to-image generation." NeurIPS. 2024.
>
> [2] Von Rütte, Dimitri, et al. "Fabric: Personalizing diffusion models with iterative feedback." arXiv preprint. 2023.
>
> [3] Radford, Alec, et al. "Learning transferable visual models from natural language supervision." ICML. 2021.
>
> [4] Li, Junnan, et al. "Blip: Bootstrapping language-image pre-training for unified vision-language understanding and generation." ICML. 2022.
>
> [5] Xu, Katherine, Lingzhi Zhang, and Jianbo Shi. "Good Seed Makes a Good Crop: Discovering Secret Seeds in Text-to-Image Diffusion Models." arXiv preprint. 2024.

---

> ### Author Response · Authors · 2024-11-25
> **Response to Reviewer Zkaj (Part 4/5)**
>
> > **W3** Concerns Regarding Human Feedback Methodology:
>
> **[Criteria for Selection]**
>
> We kindly refer the reviewer to Appendix C of our original manuscript, where we have already outlined a clear statement of the success criteria for the tasks. We believe this detailed explanation addresses the concerns raised and highlights the complexity and multi-object compositions involved in the tasks under evaluation.
>
> **[Inter-Annotator Agreement]**
>
> To address the reviewer’s concern about the lack of inter-annotator agreement and potential biases from relying on individual evaluators, we conducted an additional analysis using the “blue-rose” task (prompt: “photo of one blue rose in a vase”) as an example, given the limited time.
>
> We asked three independent evaluators to select successful images (using the criteria defined in Appendix C) from batches generated by the baselines (pretrained SD, Enhanced Prompt SD, and DreamBooth fine-tuned models) and the HERO fine-tuned model across three random seeds. While it is natural for success rates of fine-tuning-based methods to vary with seed changes, we observed that the success rates reported by different evaluators were highly consistent.
> As shown in the following table, the standard deviations in HERO’s success rates across evaluators were within 2%, demonstrating that HERO’s evaluation is robust and not significantly affected by individual biases.
> | Success Rate | seed=0 | seed=1| seed=2 |
> | --- | --- | --- | --- |
> | SD-Pretrained | 0.307 (0.045) | 0.391 (0.013) | 0.37 (0.032)  |
> | SD-Enhanced | 0.510 (0.015) | 0.474 (0.027) | 0.453 (0.022) |
> | DreamBooth | 0.354 (0.007) | 0.500 (0.056) | 0.526 (0.015) |
> | HERO (Ours) | 0.958 (0.019) | 0.818 (0.015) | 0.661 (0.015) |
>
> We hope this analysis addresses the reviewer’s concerns.
>
> > **Q1** Expand evaluation to include more diverse and complex tasks, particularly multi-object scenarios
>
> We kindly refer the reviewer to our response to W1, where we explain (1) the distinction between the number of tasks and prompts and (2) how our prompt design already incorporates complex and challenging tasks, which are addressed by prior literature (including the use of enhanced prompts in Stable Diffusion (SD), DreamBooth fine-tuning, and DDPO offline fine-tuning with large-scale pretrained models as rewards). If the reviewer has any additional prompts or tasks they would suggest we explore, we would be more than happy to include them and present the results in our camera-ready version.
>
> > **Q2** Will the authors consider performing a thorough analysis of diversity with established metrics to better understand the trade-offs in quality?
>
> We kindly direct the reviewer to our response to W2 for a detailed explanation, where we additionally examine HERO and the baselines on aesthetic scores, diversity scores, and T2I alignment scores.
>
> > **Q3** Is there a plan to establish structured protocols for feedback collection that involve multiple evaluators to enhance reliability and reduce biases?
>
> We appreciate the reviewer for proposing an interesting approach to improve our work. However, we would like to emphasize that HERO’s contribution and problem setup focus on fine-tuning Stable Diffusion (SD) to generate images that align with an individual evaluator’s intent, minimizing the need for extensive online human feedback from that specific individual. In this context, each user can personalize their fine-tuned model with HERO for their specific tasks, which may make bias reduction less relevant in this scenario. That said, we agree with the reviewer that it would be valuable to develop a more structured protocol for HERO to democratize its use and enable broader accessibility for general users. We are happy to integrate this suggestion into our plan for releasing the models and code after acceptance.

---

> ### Author Response · Authors · 2024-11-25
> **Response to Reviewer Zkaj (Part 5/5)**
>
> > **Q4** Would the authors be able to incorporate an analysis of failure cases along with strategies for mitigation in future work?
>
> We appreciate the reviewer’s suggestion regarding an analysis of failure cases and potential strategies for mitigation. We have identified two primary failure scenarios for HERO: (1) low-probability events involving the pretrained Stable Diffusion (SD), and (2) image-level failures. Below, we explain each scenario in detail and outline potential strategies for addressing them in future work:
>
> * **Low-Probability Events:**
>
> Following a common assumption in the literature, HERO also assumes that the pretrained SD can generate images that align with the given prompt with a non-zero probability. The focus of HERO is on improving the success rate rather than generating previously unattainable images. However, we have observed that when the pretrained SD generates only a few successful images that match the prompt (likely due to gaps in the SD’s training data or the imbalanced nature of web-crawled datasets [1]), HERO may struggle to effectively increase the success rate during fine-tuning. To address this, future work could explore integrating HERO with optimizing the text-embedding space for SD’s conditions or with Retrieval-Augmented Generation (RAG) to better handle low-probability events.
>
> * **Image-Level Failure:**
>
> In HERO, we use diffusion model samplers such as DDIM or DDPM with 50 denoising steps (except for in the hand anomaly correction where we use 20 steps for consistency with D3PO's setting). However, using fewer steps may result in poor image quality, which could be deemed unsuccessful based on the criteria in Appendix C. On the other hand, increasing the sampling steps can improve quality but also lengthens the training time. Balancing this trade-off is challenging, and we may explore extending HERO with distilled SD for faster generation [2].
> We would be happy to include this discussion in our revised manuscript if the reviewer suggests it.
>
> [1] Samuel, Dvir, et al. "Generating images of rare concepts using pre-trained diffusion models." AAAI. 2024.
>
> [2] Luo, Simian, et al. "Latent consistency models: Synthesizing high-resolution images with few-step inference." arXiv preprint. 2023.

---

> > ### Comment · Reviewer_Zkaj · 2024-11-28
> >
> > Thank you authors additional experimental results and clear explanations regarding the difference in DPO test-cases and HERO's. I believe the additional results presented at rebuttal, particularly the diversity and aesthetics scores, would be a great addition to the final version of the paper. I do wish that more complex task prompts can be explored, particularly ones involving more challenging compositional-scenarios, or objects with weaker priors. However, I do understand that it is time-consuming and expensive to collect human-data in short period of time. Most of my concerns have been resolved, therefore I will raise my score.

---

> > > ### Author Response · Authors · 2024-11-29
> > >
> > > We greatly appreciate the reviewer’s suggestion and constructive feedback. We have already included the results of additional metrics in Appendix F of the revised manuscript. If the reviewer has any specific prompts involving more challenging compositional scenarios or objects with weaker priors, we would be happy to test them and include the results in our final version.

---

### Official Review · Reviewer_aK5n · 2024-11-04

**Soundness:** 4
**Presentation:** 3
**Contribution:** 3
**Rating:** 6
**Confidence:** 2

**Summary:**

This paper introduces HERO, a framework for fine-tuning Stable Diffusion (SD) models using online human feedback to improve alignment with human intent. HERO addresses the limitations of traditional methods, which rely on costly predefined rewards or pre-trained models, by leveraging real-time human feedback through two main components: Feedback-Aligned Representation Learning and Feedback-Guided Image Generation. Experiments show that HERO is more efficient than prior methods in tasks such as anomaly correction, reasoning, counting, personalization, and reducing NSFW content, achieving significant improvements with minimal feedback.

**Strengths:**

* The article is well-structured and easy to follow, and the motivation is clear. The approach is simple but effective.

* The proposed method seems to be novel. And the empirical results on several tasks demonstrate the effectiveness of the proposed method.

**Weaknesses:**

* Compared to D3PO that does not require specific reward model, the proposed method in this paper clearly make the training process more complex and introduce computational overhead.

* The proposed method uses online human preferences. Does that mean the human annotator need to provide preference to the generated image at each run of the stable diffusion model? If so, it is might be difficult to collect enough data for training the encoder as contrastive learning requires a large amount of data to converge. Additionally, how to measure the performance of the trained encoder $E_\theta$?

* D3PO seems to be the closest baseline and achieves second-best results in Figure 3. Why the authors not provide results of D3PO in Table 2?

**Questions:**

Please refer to the Weakneses.

---

> ### Author Response · Authors · 2024-11-25
> **Response to Reviewer aK5n**
>
> We sincerely thank the reviewer for the thorough and constructive comments. Please find the response to your questions below.
>
> > **W1** Compared to D3PO that does not require specific reward model, the proposed method in this paper clearly make the training process more complex and introduce computational overhead.
>
> We would like to emphasize that while HERO introduces additional training for a human-aligned embedding to convert binary feedback into informative continuous reward signals, this mechanism is both efficient and effective in significantly reducing the need for online human feedback, compared to D3PO. To further illustrate the efficient training of this embedding, consider the hand deformation correction task in Figure 3. HERO requires only 1152 samples and 144 update iterations (batch size 8), compared to D3PO, which needs 5000 samples and 500 update iterations (batch size 10). Moreover, HERO's embedding map is implemented using a simple network with three CNN layers and one fully connected layer, making its training far less complex than fine-tuning Stable Diffusion.
>
> [1] Rafailov, Rafael, et al. "Direct preference optimization: Your language model is secretly a reward model." NeurIPS 2024
>
> > **W2.1** The proposed method uses online human preferences. Does that mean the human annotator need to provide preference to the generated image at each run of the stable diffusion model? If so, it is might be difficult to collect enough data for training the encoder as contrastive learning requires a large amount of data to converge.
>
> Yes. Our proposed framework HERO requires human feedback during training, similar to D3PO [1]’s setup. This setup allows for personalizing a pretrained model, such as Stable Diffusion. As pointed out by the reviewer, the bottleneck of applying this setup to a wide range of applications is the amount of human feedback required. Therefore, our motivation is to reduce the amount of online human feedback needed.
>
> We emphasize the distinction in HERO’s use of contrastive learning, which focuses on learning relationships among human-annotated samples through triplet loss. This differs from contrastive learning literature [2, 3, 4], which primarily focuses on unsupervised learning with large-scale unlabeled datasets.
>
> Specifically, HERO employs feedback-aligned representation learning by leveraging human annotations (e.g., "good," "bad," and "best") to structure embedded representations into distinct clusters using triplet loss. This enables efficient fine-tuning using continuous rewards derived from the similarity to the human-selected "best" samples. As a result, HERO significantly reduces the need for online human feedback, requiring only 0.5–1k samples, compared to baselines such as D3PO, which require at least 5k.
>
> [1] Yang, Kai, et al. "Using human feedback to fine-tune diffusion models without any reward model." CVPR 2024
>
> [2] Chen, Ting, et al. "A simple framework for contrastive learning of visual representations." ICML 2020
>
> [3] He, Kaiming, et al. "Momentum contrast for unsupervised visual representation learning." CVPR 2020
>
> [4] Caron, Mathilde, et al. "Unsupervised learning of visual features by contrasting cluster assignments." NeurIPS 2020
>
> > **W2.2** How to measure the performance of the trained encoder?
>
> Performance can be evaluated by checking whether samples labeled as positive generally receive higher rewards. For example, in the mountain task described in Table 1, we generate 128 images per epoch to learn the feedback-aligned representation. During the first epoch, the evaluator identifies 69 positive samples, but only 38 of them ranked within the top 69 rewards among all 128 samples. After 500 epochs of embedding training, 57 out of 69 positive samples achieve top-69 rewards, which improves to 63 out of 69 after 5000 training epochs.
>
> > **W3** D3PO seems to be the closest baseline and achieves second-best results in Figure 3. Why the authors not provide results of D3PO in Table 2?
>
> To ensure fair and unbiased comparisons, all our evaluations are conducted across at least three random seeds to mitigate the impact of seed variability. However, D3PO requires 5–10k feedback samples per seed for each task, which amounts to 60–120k feedback samples across the four tasks shown in Figure 3. This substantial requirement far exceeds our resource availability, making it impractical to retrain D3PO across multiple seeds for every task in our evaluation.
>
> For the hand deformation correction task highlighted in Figure 3, we instead reference the numerical results reported in their original paper. However, it is worth noting that these results are based on a single seed and lack standard deviation metrics, which limits their comparability under the robust evaluation protocol we adopted for HERO and other baselines.
>
> This resource-intensive nature of D3PO further underscores HERO’s efficiency, as HERO achieves superior performance with significantly fewer feedback samples.

---

> ### Author Response · Authors · 2024-11-29
> **A Kind Reminder for Comments**
>
> We would like to sincerely thank the reviewer for their thorough and constructive feedback. We believe that our responses adequately address the concerns raised, particularly the following points:
>
> - An explanation of HERO’s low additional training overhead, especially when compared to the costly online human feedback required by D3PO.
> - A clarification of HERO’s use of contrastive loss, which serves a different purpose than in the contrastive learning literature and does not require large amounts of data.
> - A more detailed explanation of how we measure the training of the embedding encoder.
> - A further elaboration on the extensive online human feedback requested by D3PO, which makes a direct comparison challenging.
>
> Please let us know if the reviewer has any further concerns or if additional experimental results are needed. We are committed to addressing any remaining issues, time permitting. Once again, we appreciate the reviewer’s detailed feedback and the time spent helping us improve our submission.

---

> ### Author Response · Authors · 2024-12-03
> **Kind Reminder**
>
> Dear Reviewer,
>
> The discussion phase will end in less than 9 hours.
>
> We are eager to know if we have fully addressed your concerns, as we believe we have.
> However, if you still have any questions or require further clarification, we kindly request that you leave a message before the rebuttal period ends.
> We would be more than happy to respond promptly.
>
>
> Best regards,
>
> The Authors

---

> > ### Author Response · Authors · 2024-12-04
> > **Still looking forward to the reviewer's feedback**
> >
> > Dear Reviewer,
> >
> > As the deadline for reviewers to post a message has recently passed, we understand that it is no longer possible to provide an official comment. **However, we would be deeply grateful if the reviewer could kindly edit the original review to let us know whether our rebuttal and the revised paper adequately address the questions and concerns previously raised.** Your feedback would be invaluable in helping us further improve our submission.

---

### Meta-Review · Area_Chair_7dDF · 2024-12-23

**Metareview:**

This paper looks at human feedback-guided finetuning of diffusion models, addressing various issues such as anomaly correction, reasoning, personalization, and safety. The HERO method is built on feedback-aligned representation learning and feedback-guided image generation. Experiments demonstrate more efficient and controllable generative models. Overall, reviewers appreciated the well-written paper and overall novel perspective, with concerns arising from the requirement for human preferences and collection, complexity, computational overhead introduced as a result. The authors provided a clear rebuttal including situation with respect to D3PO, mentioning that HERO requires significantly less samples and updates.Other concerns included limitations in the evaluation of the method (e.g. number of human evaluators) and evaluation of the diversity/quality/aesthetics tradeoffs. After significant back-and-forth and additional experiments for many of these concerns, all reviewers agreed that this paper warrants acceptance. After considering all of the materials, I agree with this assessment.

 I strongly suggest for the authors to include the discussions and new results (especially regarding the diversity/quality tradeoff, etc.) in the final version.

**Additional Comments On Reviewer Discussion:**

A number of concerns were raised, especially in terms of the requirements and evaluation of the method. The authors addressed these strongly, including with new experiments. The comments about the diversity/quality/aesthetics considerations are especially important and were well-addressed.

---

### Decision · Program_Chairs · 2025-01-22

Accept (Poster)